# TFMAUDIO: HIGH-FIDELITY LONG-FORM TEXT-TO-AUDIO VIA MAMBA-BASED FLOW MATCHING

## ABSTRACT

Recent advancements in audio generation have been dominated by transformer-based diffusion models, which face challenges in extrapolating positional encodings and exhibit quadratic complexity in self-attention, limiting their consistency and efficiency for long-form generation. To address these limitations, we propose TFMAudio, a novel latent audio generation model that integrates the strengths of Flow Matching and a custom-designed TFMamba backbone. TFMamba employs a dual-scan mechanism: TimeMamba captures long-range causal dependencies with linear complexity, while FrequencyMamba models spectral correlations such as harmonic structures. To enhance stability, we further introduce Energy-Aware Guidance (EAG), which mitigates state drift by adaptively regularizing classifier-free guidance. Experiments demonstrate that TFMAudio achieves state-of-the-art performance on text-to-audio benchmarks and exhibits robust extrapolation to ultra-long sequences. Remarkably, our model generates 30-minute high-fidelity audio while preserving temporal consistency and semantic alignment, significantly advancing the scalability and usability of text-to-audio models. Demo:https://huggingface.co/spaces/tfmaudio/TFMAudio

## 1 INTRODUCTION

Recent years have witnessed unprecedented advancements in generative artificial intelligence, revolutionizing domains such as text, image, video, and speech synthesis. A key driver of this progress has been the rapid evolution of diffusion models (Ho et al., 2020; Lu et al., 2022; Lipman et al., 2023; Tong et al., 2024), which have significantly propelled the state of the art across a multitude of generative tasks. While these models are most prominently celebrated for their remarkable achievements in image generation, the domain of audio generation has concurrently emerged as a field of burgeoning research interest and substantial practical impact (Kong et al., 2021; Ghosal et al., 2023; Huang et al., 2023a; Liu et al., 2023).

In practice, many contemporary Text-to-Audio (T2A) models leverage generative architectures originally conceived for the Text-to-Image (T2I) domain, such as U-Net (Ronneberger et al., 2015) / Transformers (Vaswani et al., 2017; Dosovitskiy et al., 2021; Peebles & Xie, 2023)-based diffusion backbones. These approaches have yielded remarkable results: models like AudioLDM (Liu et al., 2023; 2024), Make-an-Audio (Huang et al., 2023b;a), and Tango (Ghosal et al., 2023) demonstrate robust performance at 16kHz, while systems including Stable-Audio-Open (Evans et al., 2024), TangoFlux (Hung et al., 2024), and SoundCTM (Saito et al., 2025) deliver exceptional fidelity at a full-band(44.1kHz) sampling rate. This progress strongly suggests that the challenge of achieving high perceptual quality has been substantially addressed. Consequently, the research frontier is naturally shifting towards the synthesis of richer and more complex content, such as extended narratives or intricate musical compositions. This ambition, however, confronts the inherent redundancy of audio signals, raising a critical question: **What is the effective upper bound on the duration of consistent audio that current T2A models can generate?**

While autoregressive (AR) models are, in principle, capable of generating sequences of arbitrary length, their practical performance and perceptual quality often lag behind diffusion-based counterparts. We observe that the coherent output duration of leading T2A models is typically under 6 minutes for 16kHz audio (Liu et al., 2023; 2024; Huang et al., 2023b;a) and around 1 minute for full-band 44.1kHz audio (Evans et al., 2024; Hung et al., 2024). This fundamental limitation stems from

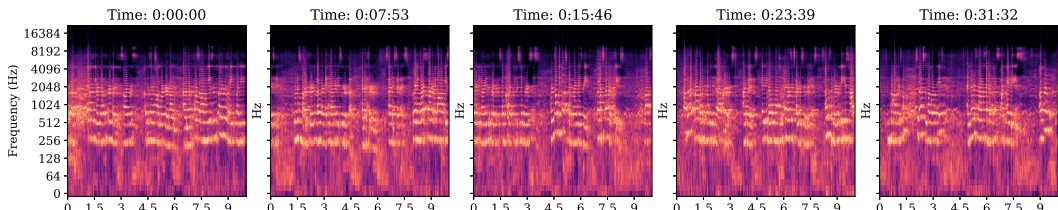

Figure 1: An example of 30 minutes audio, "a woman giving a speech in front of large audience".

the intrinsic differences between audio generation and image synthesis: (1) while images are two-dimensional spatial signals, audio is inherently a one-dimensional temporal waveform, possessing natural properties of variable length and temporal causality; (2) the information density in audio is significantly lower than in images, necessitating substantial computational resources for generation.

In contrast, the transformer architecture, which dominates current T2I / T2A designs, faces two fundamental constraints: (1) transformers lack a strong temporal inductive bias, relying instead on injected positional encodings (Dosovitskiy et al., 2021; Su et al., 2024; Kazemnejad et al., 2023) to discern sequence order; (2) the self-attention mechanism, its core component, exhibits computational and memory complexity that scales quadratically ($O(L^2)$) with the sequence length $L$. This quadratic scaling makes long-audio modeling computationally prohibitive, limiting the achievable context length and impairing generalization beyond training horizons, as well as degrading consistency and fidelity.

To address the dual challenges of efficiency and extrapolation, we introduce TFMAudio, a novel T2A architecture. Our approach pioneers the integration of a state-space model (SSM) (Gu et al., 2022) with a continuous-time generative framework for T2A. Concretely, we propose a Mamba (Gu & Dao, 2024; Dao & Gu, 2024)-based TFMamba block that jointly models time and frequency domains, trained with flow matching and augmented by energy-aware segment-wise classifier-free guidance, enabling computationally tractable, temporally consistent long-form text-to-audio synthesis. As a result, our model can generate full-band audio exceeding 30 minutes in duration from a single prompt, as shown in *Figure 1*. This capability improves the practicality of long-form audio generation, bringing resource-demanding applications such as story-length voice narration and long musical compositions within closer reach. Our contributions can be summarized as follows:

- We propose Time-Frequency Mamba, which unifies temporal causality and spectral correlation modeling via a novel dual-scan mechanism. It achieves linear complexity with respect to sequence length and leverages a recurrent hidden-state design to inherently preserve temporal consistency over long sequences.
- We introduce Energy-Aware Guidance (EAG), a stabilization strategy that decomposes the flow-matching velocity field to detect and adaptively dampen the parallel component responsible for state drift, thereby preventing numerical instability and enabling reliable ultra-long audio generation.
- We demonstrate that TFMAudio sets a new state-of-the-art on text-to-audio generation benchmarks, substantially surpassing existing models in both quality and semantic alignment. Moreover, it scales to 30-minute high-fidelity audio at 44.1 kHz with temporal consistency, markedly advancing the scalability and practical usability of text-to-audio.

## 2 RELATED WORK

**Text-to-Audio Generation.** Text-to-audio (T2A) generation has rapidly advanced with the adoption of diffusion-based latent models. Early works such as AudioLDM (Liu et al., 2023) and Make-An-Audio (Huang et al., 2023b) employed U-Net backbones at 16 kHz, demonstrating strong audio quality and text alignment. Subsequent models, including AudioGen (Kreuk et al., 2023), Tango (Ghosal et al., 2023), and Stable Audio Evans et al. (2024), extended this paradigm to 44.1 kHz (CD quality), significantly improving perceptual fidelity. To reduce sampling cost, alternatives such as consistency models (Song et al., 2023) (e.g., SoundCTM (Saito et al., 2025)) and flow matching approaches (e.g., FreeAudio (Jiang et al., 2025), TangoFlux (Hung et al., 2024)) have been introduced, enabling faster inference while maintaining quality. However, most systems remain constrained to 30s – 3mins outputs, as Transformer-style backbones suffer from quadratic

complexity and stability issues in long contexts. Achieving coherent multi-minute or tens-minute text-conditioned generation thus remains an open challenge.

**Mamba for Generative Models.** SSMs and their modern variant, Mamba (Gu & Dao, 2024; Dao & Gu, 2024), have recently demonstrated strong performance in discriminative and autoregressive tasks, as exemplified by VisionMamba (Zhu et al.) and Samba (Ren et al., 2025). Building on this momentum, several studies have begun exploring their potential in generative modeling. For instance, DiMSUM (Phung et al., 2024) integrates Mamba with Transformer in a hybrid architecture to leverage complementary strengths for diffusion-based image generation, while U-Shape Mamba (Ergasti et al., 2025) adapts SSMs into a U-Net style framework; both designs incorporate specific scanning strategies tailored for image data. In the audio domain, early efforts such as the SSM-based Sashimi (Goel et al., 2022) model operate directly on raw waveforms, although its fidelity and text alignment lag behind mainstream T2A models. Meanwhile, SpeechSSM (Park et al., 2025), designed for long-form speech synthesis, demonstrates the potential of SSMs in capturing long-range dependencies.

To the best of our knowledge, TFMAudio is the first framework that applies standalone Mamba to diffusion-based T2A generation. Unlike prior hybrid Mamba-Transformer approaches or methods that treat audio as a specialized image, TFMAudio is specifically designed to natively model the temporal and spectral characteristics of audio through its novel dual-path architecture.

## 3 PRELIMINARY

**Diffusion Model and Flow Matching.** Diffusion models formulate generation as transforming a simple noise distribution into a complex data distribution through a continuous process, typically grounded in the theory of stochastic differential equations (SDEs) or their deterministic counterparts, ordinary differential equations (ODEs). Given a data sample $\boldsymbol{x}_0 \sim q(\boldsymbol{x}_0)$, the forward process gradually perturbs it into noise by progressively injecting Gaussian noise:

$$q(\boldsymbol{x}_t|\boldsymbol{x}_{t-1}) = \mathcal{N}(\boldsymbol{x}_t; \sqrt{1-\beta_t}\boldsymbol{x}_{t-1}, \beta_t\mathbf{I})$$

The reverse process iteratively learns to denoise the corrupted samples, parameterized by a neural network $p_\theta(\boldsymbol{x}_{t-1}|\boldsymbol{x}_t)$, which approximates the score function $\nabla_{\boldsymbol{x}_t} \log p(\boldsymbol{x}_t)$ to reverse the corruption process. In contrast, flow matching methods learn a deterministic transformation from noise to data by defining a probability path between a prior distribution $p(z) = \mathcal{N}(0, \mathbf{I})$ and the data distribution $q(\boldsymbol{x}_0)$ achieved via an ODE:

$$\frac{d\boldsymbol{x}_t}{dt} = v_\theta(\boldsymbol{x}_t, t)$$

where $v_\theta(\boldsymbol{x}_t, t)$ is a time-dependent vector field learned to regresses the conditional flow $u_t(\boldsymbol{x}|\boldsymbol{z})$ from noise $\boldsymbol{z}$ to data sample $\boldsymbol{x}_0$. The training objective minimizes the difference between the model's velocity and the target vector field under the conditional probability path:

$$\mathcal{L}_{FM} = \mathbb{E}_{t,p(\boldsymbol{x}|\boldsymbol{z}),p(\boldsymbol{z})}[||v_\theta(\boldsymbol{x}_t, t) - u_t(\boldsymbol{x}_t|\boldsymbol{z})||^2]$$

Once trained, the model $v_\theta$ defines a deterministic mapping from the prior to the data distribution. To generate a sample, one first draws $\boldsymbol{z} \sim p(\boldsymbol{z})$ and then solves the probability flow ODE from $t = 0$ to $t = 1$ using any numerical ODE solver (e.g., Euler or Runge-Kutta methods).

Diffusion and flow matching models employ a parallel decoding paradigm, where the entire sequence is decoded over multiple steps. Let $L$ denote the sequence length and NFE (Number of Function Evaluations) represent the total number of sampling iterations. The computational cost can then be approximated as $c(v_\theta(L)) \times$ NFE, where $c(v_\theta(L))$ is the cost per iteration. Accordingly, efficiency can be improved along two primary directions: reducing NFE or optimizing $c(v_\theta(L))$.

**Transformer v.s. Mamba.** The transformer backbone is ubiquitous in generative modeling. A common pipeline is to patchify the input into non-overlapping patches, project each patch to a fixed-dimension embedding, add positional encodings, and model global interactions via self-attention over the resulting token sequence. While effective at capturing long-range spatial dependencies, self-attention requires pairwise interactions among all tokens and thus incurs $O(L^2)$ computation

and memory for a sequence of length L. Moreover, because self-attention is permutation-invariant, it depends on injected positional bias (e.g., RoPE) to encode order, which hampers extrapolation to longer contexts.

When applying transformers to audio latents $\boldsymbol{x} \in \mathbb{R}^{L \times C}$, conventional patchification treats the representation as an image grid and breaks the native channel-wise coupling and causal temporal structure: it disrupts cross-frequency correlations and long-range causal relationships, while the quadratic cost severely limits scalable long-form audio generation.

To enable longer and more efficient audio generation, we investigate RNN-like inductive biases, with Mamba as a promising candidate. Mamba builds upon a continuous state space representation, which is discretized for practical implementation. Its core SSM follows a recurrent formulation:

$$\boldsymbol{h}_t' = \boldsymbol{A}\boldsymbol{h}_t + \boldsymbol{B}\boldsymbol{x}_t; \quad \boldsymbol{y}_t = \boldsymbol{C}\boldsymbol{h}_t + \boldsymbol{D}\boldsymbol{x}_t$$

while a novel selection mechanism makes the transition and input matrices ($\boldsymbol{B}$ and $\boldsymbol{C}$) input-dependent, enabling context-dependent representations. Furthermore, a hardware-aware parallel algorithm ensures efficient training despite the recurrent nature of the architecture. These properties make Mamba particularly well-suited for long-form, high-fidelity audio generation: it captures long-range dependencies with linear complexity $O(L)$ in sequence length $L$, while preserving temporal coherence across extended contexts.

An inherent limitation of Mamba is its design for one-dimensional sequential data. Although raw audio is 1D, modeling waveforms at the sample level is infeasible due to extremely long sequence lengths; the dominant practice therefore encodes audio as 2D time–frequency spectrograms, which are more compact and preserve essential acoustic structure. Mamba's native 1D recurrence, however, is not well matched to spectral patterns and cross-channel correlations. To bridge this gap and leverage Mamba's efficiency for long-range dependencies, we introduce a novel Mamba-based block tailored to operate on 2D spectrogram representations.

# 4 TFMAudio

We design TFMAudio (Time-Frequency Mamba-based Audio) following the latent diffusion paradigm, where audio is first mapped into a compact latent space before generative modeling. In this section, we will introduce the core of our backbone, TFMamba, and an energy-aware guidance strategy during TFMAudio sampling.

## 4.1 Time-Frequency Mamba Block

State-of-the-art generative models for audio typically encode signals into a two-dimensional latent representation of shape $(L, C)$, where $L$ denotes length and $C$ denotes channels. For instance, Stable Audio Open (Evans et al., 2024) uses a VAE to encode audio into a 64-channel 2D latent, which is then patchified and processed with positional encodings by a transformer, treating it as an image. Similarly, Mamba-based diffusion models, such as DiMSUM (Phung et al., 2024), employ specialized scanning strategies to extend the 1D-oriented Mamba to handle 2D images.

However, audio latents fundamentally differ from image latents. In particular, the temporal axis of audio latents is separable: a latent of shape $(L, C)$ can be partitioned into multiple contiguous segments along the length dimension:

$$(L, C) = \{(L_{sub1}, C), (L_{sub2}, C), \dots (L_{subk}, C)\}; L = L_{sub1} + L_{sub2} + \cdots + L_{subk}$$

each of which can be independently decoded into a coherent audio clip. This property implies that conventional patchification, which treats the latent as a grid of independent patches, disrupts the inherent channel-wise coupling in audio representations and discards essential long-range causal dependencies across time. This motivates us to design a spectrogram-oriented Mamba block that respects temporal separability while maintaining efficiency.

**Time Mamba.** Intuitively, audio can be treated as sequential data and modeled directly using Mamba. In our proposed TimeMamba block, the audio latent representation is processed as a sequence of shape $(1, C)$ along the temporal dimension. As illustrated in *Figure 2(right)*, the model scans the sequence from left to right, updating a hidden state $\boldsymbol{h}_t$ at each step to predict the subsequent representation. This design offers several key advantages:

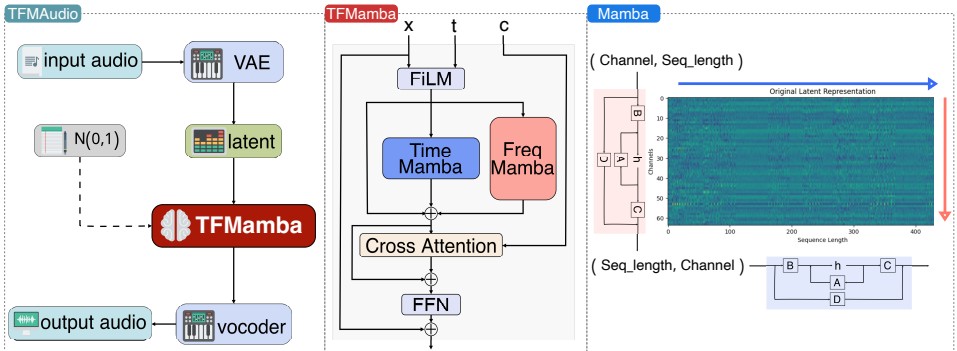

Figure 2: The key components of TFMAudio: (left) the workflow of Latent Flow Matching, (middle) the architecture of the TFMamba block, and (right) the dual-scanning mechanism of Mamba.

- Inherent Temporal Modeling: The RNN-like recurrent state update naturally preserves chronological relationships and causal structure, which is fundamental to audio signals.
- Physically-Grounded Dynamics: SSMs are inherently designed to describe dynamical systems. Since audio is itself a physical phenomenon generated by a dynamical process, this formulation offers a more natural inductive bias for audio generation.
- Linear Computational Efficiency: By replacing the quadratic complexity of self-attention, TimeMamba achieves $O(L)$ complexity with respect to sequence length, enabling scalable long-form audio generation.

Despite TimeMamba's strong capability in temporal modeling, its key shortcoming is the limited ability to capture coherent spectral structures, such as harmonic resonances across frequencies. Audio signals exhibit intricate dependencies between frequency components that do not conform to strictly sequential patterns. SSMs are inherently less effective at capturing non-causal, synchronous relationships across channels (frequencies).

**Frequency Mamba.** To overcome the limitation in spectral resonance modeling, we introduce FrequencyMamba, a variant that reinterprets the spectrogram latent as a sequence along the frequency channel axis. FrequencyMamba scans the input from the highest to the lowest channel, which means the input shape is $(1, L)$, thereby capturing cross-frequency correlations and harmonic structures. Since scanning across $C$ requires a fixed length, which limits its ability to process audio of arbitrary length, we exploit the temporal separability of audio and instead scan shorter subsequences $(1, L_{sub})$, enabling variable-length generation while preserving local harmonic relationships.

The dual-scan mechanism is illustrated in *Figure 2(right)*: TimeMamba operates along the temporal dimension to capture long-range causal dependencies, while FrequencyMamba scans along the frequency axis to model spectral correlations within local segments.

**Conditional Generation.** Conditioning signals guide the model toward semantically aligned outputs. Following the standard formulation of conditional diffusion models, the network input is represented as a triplet $(\boldsymbol{x}_t, t, \boldsymbol{c})$, where $\boldsymbol{x}_t$ is the noisy latent at timestep $t$, $t$ is the diffusion timestep embedding, and $\boldsymbol{c}$ denotes the conditioning signal (e.g., text prompt embeddings).

For temporal conditioning, we adopt Feature-wise Linear Modulation (FiLM) (Perez et al., 2018). The timestep $t$ is projected via a learned embedding function into affine parameters $(\gamma_t, \beta_t)$, which are then applied within each TFMamba block to modulate activations:

$$\text{FiLM}(\boldsymbol{x}_t, t) = \gamma_t \odot \boldsymbol{x}_t + \beta_t$$

For semantic conditioning, we incorporate cross-attention to align the noisy latent representation with the conditioning signal. The cross-attention mechanism is defined as:

$$\text{CrossAttn}(Q, K, V) = \text{Softmax}(\frac{QK^T}{\sqrt{d}})V$$

where $Q = W_Q \cdot \boldsymbol{x}_t, K = W_K \cdot \boldsymbol{c}, V = W_V \cdot \boldsymbol{c}$, and $d$ is the feature dimension for scaling. This design enables the model to dynamically attend to the most semantically relevant components of c, ensuring faithful alignment between conditioning inputs and generated audio.

Building on the above, we propose the TFMamba block, as shown in *Figure 2(middle)*. Unlike hybrid approaches that simply combine Mamba with Transformers, TFMamba's native dual-path design jointly captures temporal causality and spectral resonance with linear complexity, ensuring both scalability and fidelity in long-form audio generation.

**Computational Complexity Analysis.** For TimeMamba and FrequencyMamba, both components build upon the SSM, whose complexity scales linearly with sequence length. For an input spectrogram latent $\boldsymbol{x} \in \mathbb{R}^{L \times C}$, the complexity of applying SSM updates is $O(L \cdot C)$. For FiLM Modulation, it introduces minimal overhead, requiring only element-wise affine transformations. Therefore, the FiLM operation has complexity $O(L \cdot C)$, maintaining the linear scaling characteristics of the overall architecture. For Cross-Attention: While cross-attention between the latent sequence $(L, C)$ and conditioning embeddings $(S, C)$ has complexity $O(L \cdot S \cdot C)$, where $S$ is the length of the conditioning sequence, we note that in practice $S$ is fixed and $\ll L$ for typical audio generation scenarios.

The overall complexity of TFMamba can be summarized as: $O(L \cdot C + L \cdot S \cdot C) \approx O(L)$ assuming $C$ and $S$ are constants or significantly smaller than $L$. This linear scaling enables efficient processing of long audio sequences while maintaining strong modeling capabilities.

**Flow Matching Training.** We adopt the Optimal Transport Flow Matching (OT-FM) framework (Tong et al., 2024) to train our generative model. OT-FM provides a simulation-free training objective, avoiding costly stochastic trajectory sampling, and instead relies on analytically defined optimal transport paths for efficient and stable learning. Formally, let $\boldsymbol{x}_1 \sim \mathcal{N}(0, \mathbf{I})$ denote the initial noise distribution and $\boldsymbol{x}_0 \sim q(\boldsymbol{x})$ be the target audio latent representation. For each timestep $t \in [0, 1]$, we define the linear interpolation path:

$$\boldsymbol{x}_t = t\boldsymbol{x}_0 + (1 - t)\boldsymbol{x}_1$$

Our parameterized vector field $v_\theta(\boldsymbol{x}_t, t, \boldsymbol{c})$, conditioned on timestep $t$ and semantic signal $\boldsymbol{c}$, is trained to approximate this velocity field $(\boldsymbol{x}_0 - \boldsymbol{x}_1)$. The Flow Matching loss is therefore:

$$\mathcal{L}_{FM} = \mathbb{E}_{\boldsymbol{x}_1 \sim \mathcal{N}(0,\mathbf{I}), \boldsymbol{x}_0 \sim q(\boldsymbol{x}), t \sim \mathcal{U}[0,1]}[||v_\theta(\boldsymbol{x}_t, t, \boldsymbol{c}) - (\boldsymbol{x}_0 - \boldsymbol{x}_1)||^2]$$

## 4.2 ENERGY-AWARE ADAPTIVE GUIDANCE

To further improve sample quality and strengthen alignment with conditioning signals during inference, we incorporate Classifier-Free Guidance (CFG) (Ho & Salimans, 2021). Within the flow matching framework, CFG can be interpreted as steering the predicted vector field toward the conditional direction while simultaneously repelling it from the unconditional counterpart.

Formally, let $\omega > 1$ denote the guidance scale. The guided vector field $\hat{v}_\theta(\boldsymbol{x}_t, t, \boldsymbol{c})$ is computed as:

$$\hat{v}_\theta(\boldsymbol{x}_t, t, \boldsymbol{c}) = v_\theta(\boldsymbol{x}_t, t, \emptyset) + \omega\boldsymbol{r}_t; \quad \boldsymbol{r}_t = v_\theta(\boldsymbol{x}_t, t, \boldsymbol{c}) - v_\theta(\boldsymbol{x}_t, t, \emptyset)$$

where $v_\theta(\boldsymbol{x}_t, t, \emptyset)$ is the unconditional prediction obtained by replacing $\boldsymbol{c}$ with a null token $\emptyset$, $\boldsymbol{r}_t$ is the residual velocity.

While the introduction of a hidden state via TFMamba equips the model with temporal causality, it also introduces the issue of state drift. Particularly during long-sequence extrapolation, the accumulation of state errors, exacerbated by out-of-distribution (OOD) inputs, can cause the generated trajectory to diverge. When $\omega > 1$, the residual $\boldsymbol{r}_t$ may push $\boldsymbol{x}_t$ away from the high-likelihood manifold into high-norm regions. This deviation renders the underlying ODE system stiff, where incremental errors accumulate and diverge rapidly, ultimately leading to the structural collapse of the generated audio.

Let $\Delta$ denote the step size of the ODE solver, the growth in the norm of $x_t$ is then given by:

$$||\boldsymbol{x}_{t+\Delta}||^2 - ||\boldsymbol{x}_t||^2 = 2\Delta(\boldsymbol{x}_t \cdot \hat{v}_\theta(\boldsymbol{x}_t, t, \boldsymbol{c})) + \Delta^2 ||\hat{v}_\theta(\boldsymbol{x}_t, t, \boldsymbol{c})||^2$$

Consequently, controlling the norm of $\boldsymbol{x}_t$ requires imposing constraints on the velocity field $||\hat{v}_\theta(\boldsymbol{x}_t, t, \boldsymbol{c})||^2$. We observe that the unconditional velocity is more stable, as it predicts a path near the latent mean, whereas the conditional velocity is less stable. To isolate the fast dynamics, we project the residual vector onto the direction of the conditional guidance:

$$\boldsymbol{r}_t^{\parallel} = \frac{\boldsymbol{r}_t \cdot v_\theta(\boldsymbol{x}_t, t, \boldsymbol{c})}{||v_\theta(\boldsymbol{x}_t, t, \boldsymbol{c})||^2 + \epsilon} v_\theta(\boldsymbol{x}_t, t, \boldsymbol{c}), \quad \boldsymbol{r}_t^{\perp} = \boldsymbol{r}_t - \boldsymbol{r}_t^{\parallel}$$

Thus, our objective is to prevent the norm of the parallel component from becoming excessively large. We define this norm $||\boldsymbol{r}_t^{\parallel}||^2$ as the "energy" $\mathcal{E}$ of the update. As our analysis indicates, divergence typically occurs during the generation of long sequences under OOD conditions. To proactively identify these potential failure segments, we detect unstable segments by comparing the log-energy of each segment to the robust median of the sequence's log-energies. A segment is flagged if $\log(\mathcal{E}_{sub}) - \log \mathcal{E}_{median} > \text{tol}$. As a preventative measure, we attenuate the guidance scale $\omega$ specifically within these detected segments to prevent the structural collapse of generated samples:

$$\omega'_{sub} = \text{clip}(\sqrt{\frac{\mathcal{E}_{median}}{\mathcal{E}_{sub}}}, \delta, 1) \cdot \omega$$

We term this the Energy-Aware Adaptive Guidance (EAG) mechanism. By selectively adjusting only the segments prone to norm explosion, EAG mitigates the drift of $||\boldsymbol{x}_t||^2$ while preserving the quality in stable regions. In contrast to a standard CFG, which treats the guidance scale $\omega$ as a global scalar, our EAG framework extends $\omega$ to a duration-varying vector $\vec{\omega}$. This provides fine-grained, segment-level control along the temporal axis, which aligns better with the time-varying nature of audio signals. More details can be found in the Appendix A.3.

## 5 EXPERIMENTS

Following established practices in latent audio models, we adopt the VAE from Stable Audio Open as our audio encoder and utilize the T5 (Raffel et al., 2020) model as our text prompt encoder. The backbone of TFMAudio is constructed by stacking multiple TFMamba blocks, with detailed architectural parameters provided in the Appendix A.1.

### 5.1 EXPERIMENTAL SETUP

**Datasets.** We trained our model using the AudioCaps (Kim et al., 2019) and WavCaps (Mei et al., 2024) training sets. All audio clips were resampled to 44.1 kHz and clipped to 10 seconds. For testing, we evaluated our model on the AudioCaps full test set, which were also resampled to 44.1 kHz.

**Training Details.** We trained our model using 8 NVIDIA A100 GPUs with a batch size of 64 per GPU for 100 epochs. The AdamW optimizer was employed with an initial learning rate of $1 \times 10^{-4}$, decayed using a cosine annealing scheduler over two decay cycles. During sampling, the CFG scale $\omega$ was set to 2.5, and the decay factor $\delta$ in our energy-aware guidance (EAG) strategy was set to 0.8.

**Baselines.** We compared our model against three state-of-the-art full band (44.1 khz) text-to-audio generation models: AudioLDM2-48K (Liu et al., 2024), Stable Audio Open (Evans et al., 2024), and TangoFlux (Hung et al., 2024). For each baseline, we utilized the officially released pre-trained checkpoints and adhered to their recommended configurations to report their best performance.

**Evaluation Metrics.** To comprehensively evaluate the generated audio, we employ four established metrics covering both intrinsic audio quality and text-audio alignment: CLAP (Wu* et al., 2023) Score (text alignment), PANNs (Kong et al., 2020)-based Inception Score (quality/diversity), VGGish (Hershey et al., 2017)-based Frechet Audio Distance (FAD) (Kilgour et al., 2019) (quality/distributional match), and PANNs-based KL divergence (distributional match).

### 5.2 RESULTS

**Main Results.** To assess the model's learning and extrapolation capabilities, we generated audio samples of varying durations (10s, 30s, and 60s). Notably, TangoFlux was trained on 30s clips and is constrained to a maximum generation length of 30s, while Stable Audio Open supports up to 47s. Table 1 summarizes the main results, showing that TFMAudio demonstrates substantially stronger performance: it achieves significantly higher CLAP scores than all baselines and establishes new state-of-the-art results for both 30s and 60s generations. Importantly, as generation length increases, TFMAudio exhibits only marginal degradation, underscoring its superior stability and robust extrapolation capability relative to existing models.

Table 1: Main Results

| Duration | Model | Parameters↓ | NFE↓ | CLAP↑ | IS↑ | FAD↓ | FD↓ | KL↓ |
|---|---|---|---|---|---|---|---|---|
| 10s | AudioGen (16K) | 1.5B | - | 0.2730 | 6.8701 | 6.7073 | 36.1438 | 2.5361 |
| | Make-an-Audio2 (16K) | 937M | 100 | 0.2987 | 9.1671 | **1.0705** | 11.3228 | 1.1418 |
| | EzAudio (16K) | 874M | 50 | 0.3634 | 11.0158 | 2.9104 | **10.5041** | 1.1490 |
| | IMPACT (16K)* | 427M | - | 0.364 | 10.53 | 1.17 | 14.72 | **1.07** |
| | AudioLDM2-48K | 1.7B | 200 | 0.5940 | 5.4737 | 4.3070 | 27.6577 | 1.7278 |
| | Stable Audio Open | 1B | 100 | 0.3873 | 9.3759 | 3.4959 | 28.1434 | 1.9889 |
| | TangoFlux | **500M** | 50 | 0.6513 | **12.4480** | 2.2147 | 16.4745 | 1.1006 |
| | TFMAudio | 581M | **20** | **0.6684** | 11.8419 | 2.7874 | 14.4348 | 1.0996 |
| 30s | AudioLDM2-48K | 1.7B | 200 | 0.4687 | 4.7362 | 8.9574 | 43.1640 | 2.0527 |
| | Stable Audio Open | 1B | 100 | 0.3787 | 9.3843 | 4.1255 | 24.7685 | 1.8984 |
| | TangoFlux | **500M** | 50 | 0.5548 | 10.5810 | 2.9025 | **19.4439** | **1.110** |
| | TFMAudio | 581M | **20** | **0.6230** | **12.3884** | **2.7416** | 23.7232 | 1.1681 |
| 60s | AudioLDM2-48K | 1.7B | 200 | 0.4334 | 4.0597 | 10.5725 | 55.5130 | 2.2770 |
| | Stable Audio Open(47s) | 1B | 100 | 0.3669 | 9.3531 | 4.3031 | **26.3675** | 1.8482 |
| | TangoFlux | - | - | - | - | - | - | - |
| | TFMAudio | **581M** | **20** | **0.6275** | **12.1987** | **2.7304** | 32.0850 | **1.2168** |

⋆ The results are from the IMPACT paper (Huang et al., 2025).

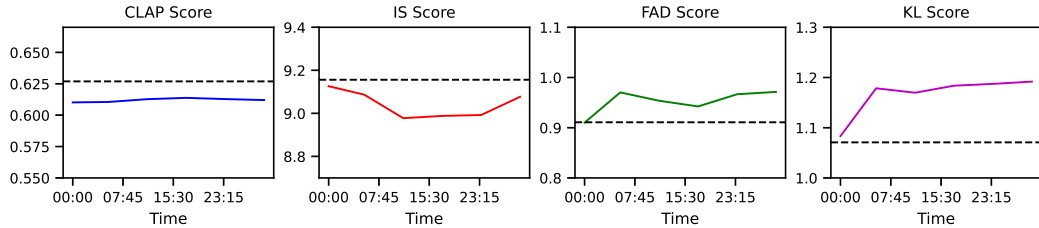

Figure 3: The quality of the generated audio varies over time, x-axis is the duration of audio.

**Ultra Long Audio Generation.** To further assess the extrapolation capability of TFMAudio, we conducted an ultra-long generation experiment, producing a continuous 30-minute audio sequence. To evaluate the quality and consistency of such long-form generation, we extracted multiple non-overlapping 30-second clips from different temporal segments and computed the same four evaluation metrics against the test set. This protocol enables a fine-grained assessment of temporal consistency over extended durations.

The results are reported in *Figure 3*, where the black dashed line denotes the reference performance of standard 30s generations by TFMAudio. As anticipated, we observe a mild degradation in all metrics over the 30-minute sequences. This is expected, as prompts requiring temporal coherence naturally exhibit greater variation in longer contexts. Despite this, the overall performance remains remarkably stable. The metric distributions across all evaluated clips (CLAP: $0.6120 \pm 0.0014$, IS: $9.0413 \pm 0.0625$, FAD: $0.9523 \pm 0.0235$, KL: $1.1657 \pm 0.0411$) consistently approach state-of-the-art (SOTA) levels, with no clip showing catastrophic failure. These results demonstrate that TFMAudio not only excels in short-form synthesis but also effectively maintains semantic coherence and acoustic fidelity for ultra-long audio generation.

**Sampling Efficiency.** We empirically validated the linear complexity of TFMamba by benchmarking the sampling efficiency of TFMAudio against a Transformer-based flow matching model, measuring inference latency across varying output lengths. To ensure a fair comparison, both models use the identical ODE solver with the same number of function evaluations (NFE). As shown in *Figure 4*, the sampling time of TFMamba scales approximately linearly with sequence length, while the transformer-based baseline exhibits a clear super-linear (nearly quadratic) scaling trend. These results provide robust empirical evidence for TFMamba's linear scaling property, underscoring its computational advantage and scalability for long-form audio generation.

**Ablation Study** We conducted ablation studies on TFMAudio to validate the efficacy of its key components: the FrequencyMamba module and the EAG strategy. In addition, we trained a 1B-

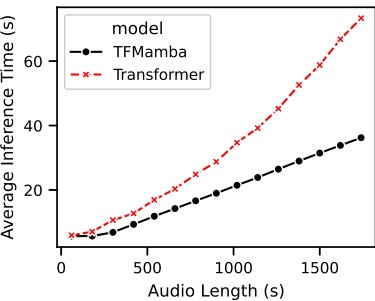

Figure 4: The generated time varies with the length of the audio.

Table 2: Ablation results

| Duration | Model | Parameters↓ | CLAP↑ | IS↑ | FAD↓ | FD↓ | KL↓ |
|---|---|---|---|---|---|---|---|
| 10s | w/o FreqMamba | 579M | 0.5784 | 10.0166 | 6.9879 | 18.1077 | 1.1983 |
| | Transformer | 1B | 0.6421 | 9.9798 | 3.7796 | 23.4039 | 1.1358 |
| | w/o EAG | 581M | 0.6672 | 11.8419 | 2.7978 | 14.4588 | 1.1030 |
| | TFMAudio | 581M | **0.6684** | **11.8419** | **2.7874** | **14.4348** | **1.0996** |
| 30s | w/o FreqMamba | 579M | 0.3085 | 5.5623 | 9.0628 | 45.6336 | 1.3096 |
| | Transformer | 1B | 0.6119 | 10.4299 | 3.4642 | 25.4411 | 1.2142 |
| | w/o EAG | 581M | 0.6270 | 12.3790 | 2.7432 | 23.7843 | 1.1694 |
| | TFMAudio | 581M | **0.6230** | **12.3884** | **2.7416** | **23.7232** | **1.1681** |
| 60s | w/o FreqMamba | 579M | 0.1999 | 4.6078 | 11.4394 | 59.1374 | 1.4570 |
| | Transformer | 1B | 0.5843 | 9.3891 | 3.7796 | **27.9194** | 1.2979 |
| | w/o EAG | 581M | 0.6272 | 12.2192 | 2.7310 | 32.2067 | 1.2182 |
| | TFMAudio | 581M | **0.6275** | **12.1987** | **2.7304** | 32.0850 | **1.2168** |

parameter Transformer (following the architecture used in Stable Audio Open) under the same flow-matching framework. This controlled experiment helps isolate whether the performance improvement stems primarily from the flow-matching objective or from the TFMamba architecture itself.

Table 2 presents the ablation results. Removing the FrequencyMamba module (reducing the model to a vanilla Mamba backbone) leads to a substantial performance decline across all metrics, most notably in CLAP, IS, and FAD, despite only a minor reduction of 2M parameters. This highlights the indispensable role of FrequencyMamba in modeling spectral correlations and harmonic structures that are critical for high-fidelity audio generation.

In addition, while the flow-matching-trained Transformer achieves competitive performance at short durations (10s), its quality deteriorates sharply when generating longer sequences. This observation indicates that the strong extrapolation capability is not attributable to the flow-matching objective itself, but rather to the inherent property of the TFMamba block. Finally, disabling EAG results in a consistent performance degradation. Notably, this improvement is readily observable even at a low guidance scale (CFG=2.5), which minimizes the occurrence of high-energy segments. The consistent gains confirm EAG's utility as a robust stabilization mechanism. Extended analysis is in Appendix A.3.

# 6 CONCLUSION AND DISCUSSION

We presented TFMAudio, a latent flow-matching based text-to-audio framework that combines the efficiency of state-space models with the spectral modeling power required for high-quality audio synthesis. By unifying TimeMamba and FrequencyMamba in a dual-scan architecture, and introducing Energy-Aware Guidance to stabilize long sequence generation, TFMAudio overcomes the fundamental limitations of transformer-based diffusion models. Our approach not only achieves state-of-the-art results on standard text-to-audio benchmarks but also enables the generation of ultra-long (30+ minutes) high-fidelity audio with strong temporal consistency.

Furthermore, by virtue of Mamba's inherent support for RNN-style inference, TFMAudio holds the potential to operate efficiently on resource-constrained devices. Its recurrent hidden state mechanism

also provides a natural pathway toward autoregressive generation, opening the door to integration with large language models for unified and scalable multimodal systems. We believe that TFMAudio not only creates new opportunities for practical long-form audio applications—such as story narration, music composition, and immersive media generation—but also points to a promising direction for efficient and scalable multimodal sequence modeling.

## REPRODUCIBILITY STATEMENT

To ensure reproducibility, our study adheres to the following protocols: all experiments utilize publicly available datasets, our implementation is built upon open-source libraries, and we employ a fixed random seed across all trials.

Moreover, we have provided comprehensive implementation details in the Appendix, including model configurations (Appendix A.1), training and sampling procedures (Appendix A.2), and hyperparameter settings (Appendix A.3). We will also release our codebase and pre-trained models upon publication to facilitate further research and validation by the community.

## ETHICS STATEMENT

This work is a research study that utilizes exclusively publicly available datasets and open-source models. All materials are used in strict accordance with their respective licenses and for non-commercial research purposes only. We confirm that the data contains no personal identifiable information, and the research does not involve the creation of harmful content. As such, this study raises no ethical concerns.

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

# A    APPENDIX

## USE OF LARGE LANGUAGE MODELS

During the preparation of this paper, the authors used Large Language Models (LLMs) only for the purpose of language polishing and refinement. After using LLMs, the authors reviewed and edited the content as needed and take full responsibility for the content of the publication.

## A.1    MODEL CONFIGURATIONS

TFM-Audio follows a latent diffusion paradigm with the following components:

**Audio VAE.**   We employ the VAE from Stable-Audio-Open as our latent audio encoder and vocoder. This VAE features a 64-channel bottleneck and achieves a compression factor of 512. For a 44.1 kHz audio signal, the encoder produces a latent representation of shape $(L, 64)$, where the sequence length $L$ is calculated as $L = \text{duration} \times 44100/512$.

**Text Encoder.**   We utilize the T5-base model as our text encoder. For each input prompt, the encoder produces a fixed-dimensional representation with a shape of $(128, 768)$, corresponding to 128 token embeddings each of dimensionality 768.

**Time and Frequency Mamba.**   Our TimeMamba module is implemented based on the official Mamba2 (Dao & Gu, 2024) source code (`https://github.com/state-spaces/mamba.git`), with a model dimension of 768 and a state dimension of 128. Similarly, FrequencyMamba follows the same implementation but with a model dimension of 64 while maintaining the same state dimension of 128.

**FiLM.**   The timestep embedding is generated using a sinusoidal positional encoding, which is then projected through two linear layers with a GELU activation in between to produce the parameters $(\gamma_t, \beta_t)$ with shape (1, 768).

**Cross-Attention.**   The cross-attention mechanism employs 16 attention heads, and the total model dimension is 768. The key and value projections are derived from the T5 text embeddings, while the query projection is obtained from the latent representation.

**Overall Architecture.**   The overall architecture begins with an initial linear projection layer that maps the 64-channel latent input to a 768-dimensional feature space. This is followed by a stack of 48 identical TFMamba modules. Each TFMamba module contains a TimeMamba block and a FrequencyMamba block, augmented with FiLM and cross-attention. The network concludes with a final linear projection layer that maps the 768-dimensional features back to the original 64-channel latent space.

## A.2    TRAINING AND SAMPLING DETAILS

The training procedure, outlined in *Algorithm 1*, minimizes the loss function depicted in *Figure 5*. The training curve exhibits a slight discontinuity, which aligns with the scheduled restart of the cosine annealing learning rate.

**Algorithm 1:** Flow Matching Training

**Input** : datasets
$(q(\boldsymbol{x}), \boldsymbol{c}), v_\theta, t_{scheduler}, \epsilon = 0.1$

1 **for** *epoch in epochs* **do**
2    // Sample data
3    Sample $\boldsymbol{x}_0 \sim q(\boldsymbol{x}), \boldsymbol{x}_1 \sim \mathcal{N}(0, \mathbf{I}), t \sim t_{scheduler}$
4    **if** $random() < \epsilon$ **then**
5      | $\boldsymbol{c}_{emb} \leftarrow \emptyset$
6    **end if**
7    **else**
8      | $\boldsymbol{c}_{emb} \leftarrow$ Encoder($\boldsymbol{c}$)
9    **end if**
10   // Interpolate
11   $\boldsymbol{x}_t \leftarrow t\boldsymbol{x}_0 + (1-t)\boldsymbol{x}_1$
12   // Predict velocity
13   $\hat{v}_\theta \leftarrow v_\theta(\boldsymbol{x}_t, t, c_{emb})$
14   // Compute loss
15   $\mathcal{L}_{FM} \leftarrow ||\hat{v}_\theta - (\boldsymbol{x}_0 - \boldsymbol{x}_1)||^2$
16   Update $v_\theta$ using AdamW optimizer
17 **end for**

**Algorithm 2:** TFMAudio Sampling

**Input** : $v_\theta, \omega, \delta$, prompt, step-size=0.05, tol=0.05

1 $\boldsymbol{x}_t \sim \mathcal{N}(0, \mathbf{I})$
2 **for** *t in [0, 1] with step-size* **do**
3    $\boldsymbol{c}_{emb} \leftarrow$ Encoder(prompt)
4    $\boldsymbol{v}^{cond} \leftarrow v_\theta(\boldsymbol{x}_t, t, c_{emb})$
5    $\boldsymbol{v}^{uncond} \leftarrow v_\theta(\boldsymbol{x}_t, t, \emptyset)$
6    $\boldsymbol{r} \leftarrow \boldsymbol{v}^{cond} - \boldsymbol{v}^{uncond}$
7    $\boldsymbol{r}^{\|} \leftarrow \frac{\boldsymbol{r} \cdot \boldsymbol{v}^{cond}}{||\boldsymbol{v}^{cond}||^2 + \epsilon} \boldsymbol{v}^{cond}$
8    $\mathcal{E}_{median} \leftarrow$ Median($[||\boldsymbol{r}^{\|}_{sub}||^2]$)
9    **for** *each segment $L_{sub}$* **do**
10      **if** $\log(\mathcal{E}_{sub}) - \log(\mathcal{E}_{median}) > tol$ **then**
11        | $\boldsymbol{v}'_{sub} \leftarrow \boldsymbol{v}^{uncond}_{sub} +$ clip($\sqrt{\frac{\mathcal{E}_{median}}{\mathcal{E}_{sub}}}, \delta, 1) \cdot \omega \cdot \boldsymbol{r}_{sub}$
12      **end if**
13      **else**
14        | $\boldsymbol{v}'_{sub} \leftarrow \boldsymbol{v}^{uncond}_{sub} + \omega \cdot \boldsymbol{r}_{sub}$
15      **end if**
16    **end for**
17    $\boldsymbol{x}_t \leftarrow \boldsymbol{x}_t +$ step-size $\cdot [\boldsymbol{v}'_{sub}]$
18 **end for**

Furthermore, *Figure 5* compares the training loss against an ablation variant where the Frequency-Mamba module is removed. The results demonstrate that our full model converges both faster and to a lower loss, underscoring the critical contribution of the FrequencyMamba module to training stability and efficiency.

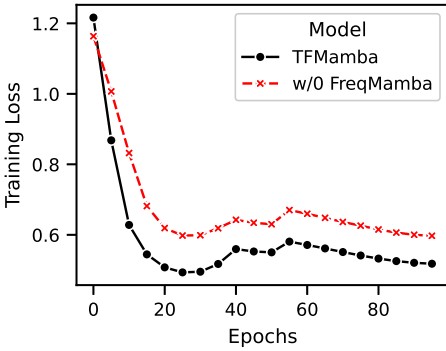

Figure 5: The loss curve during training.

During sampling, as detailed in *Algorithm 2*, we employ an Euler solver with a fixed step size of 0.05. For the energy-aware adaptive guidance (EAG) mechanism, the guidance scale $\omega$ is set to 2.5, the tolerance threshold is set to 0.05, and the decay factor $\delta$ is set to 0.8.

### A.3 ANALYSIS OF HYPERPARAMETER SENSITIVITY

**CFG Scale Analysis** Our experiments demonstrate that the classifier-free guidance scale ($\omega$) significantly influences objective audio quality metrics (CLAP, IS, FAD, KL). Crucially, we find this relationship to be non-monotonic, complicating the identification of a single globally optimal $\omega$. We argue this intricacy stems from the fact that optimal noise-fidelity trade-offs are context-dependent,

varying across audio clips and generation lengths. This observation directly motivates our proposed approach of employing a segment-adaptive, duration-varying $\omega$.

Table 3: Performance varies with $\omega$

| Duration | $\omega$ | CLAP↑ | IS↑ | FAD↓ | KL↓ |
|---|---|---|---|---|---|
| 10s | 4.5 | 0.6564 | 8.5374 | 0.9827 | 1.0296 |
| | 3.5 | 0.6631 | 8.5994 | 0.9160 | 1.0163 |
| | 2.5 | 0.6690 | **8.6248** | 0.8756 | 0.9965 |
| | 1.5 | **0.6715** | 8.4683 | **0.8641** | **0.9768** |
| 30s | 4.5 | 0.6202 | 9.1372 | 0.9627 | 1.1134 |
| | 3.5 | 0.6234 | **9.2094** | 0.9138 | 1.0959 |
| | 2.5 | 0.6235 | 9.1701 | **0.9010** | 1.0713 |
| | 1.5 | **0.6259** | 8.9013 | 0.9296 | **1.0528** |
| 60s | 4.5 | 0.6236 | 9.1042 | 0.9707 | 1.1839 |
| | 3.5 | 0.6268 | **9.1597** | 0.9327 | 1.1585 |
| | 2.5 | **0.6272** | 9.1096 | **0.9235** | 1.1314 |
| | 1.5 | 0.6226 | 8.7939 | 0.9554 | **1.0875** |

We extended our analysis of the guidance scale $\omega$ to ultra-long audio generation. While the objective quality metrics remain significantly affected by $\omega$, as observed with short audio, its influence on output consistency—measured by the variance across temporal segments—is negligible. This divergence demonstrates that TFMAudio exhibits considerable robustness to the CFG scale in terms of maintaining consistent output over long sequences.

Table 4: Ultra-long audio performance varies with $\omega$

| $\omega$ | CLAP↑ | IS↑ | FAD↓ | KL↓ |
|---|---|---|---|---|
| 4.5 | $0.6081 \pm 0.0021$ | $\mathbf{9.1617 \pm 0.0351}$ | $1.0096 \pm 0.0233$ | $1.1990 \pm 0.0375$ |
| 2.5 | $\mathbf{0.6120 \pm 0.0014}$ | $9.0413 \pm 0.0625$ | $\mathbf{0.9523 \pm 0.0235}$ | $\mathbf{1.1657 \pm 0.0411}$ |

**EAG Decay Factor Analysis** Since TFMAudio was trained on 10-second audio clips, its extrapolation to longer sequences (e.g., 30s or 60s) may occasionally exhibit numerical instability, manifesting as audible artifacts or discontinuities (see *Figure 6*).

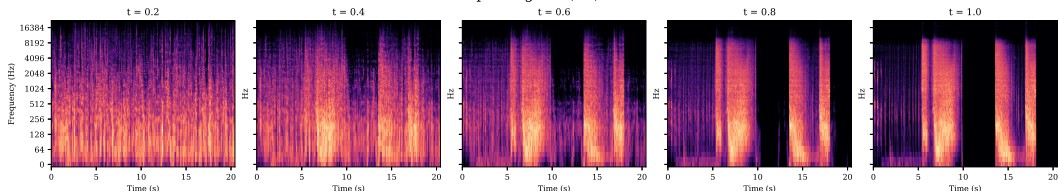

Figure 6: Unnatural discontinuity after 10 seconds.

We identify the root cause as uncontrolled velocity fields when the model operates outside the training distribution. Adaptive Projected Guidance (APG) (Sadat et al., 2025a;b), an adaptive CFG method designed for image generation, prevents oversaturation by suppressing parallel components. While effective in its original domain, we empirically show that this global adjustment of velocity fields leads to degraded quality in audio generation, as quantified in *Table 5*. This limitation arises because a global suppression strategy is too coarse for the long-term temporal dependencies in audio. Consequently, we propose a segment-wise guidance modulation strategy that precisely controls the guidance strength locally to maintain stability without compromising perceptual quality.

We also conducted a hyperparameter analysis of EAG strategy. While EAG might appear to simply achieve performance gains by locally adjusting the guidance scale $\omega$, we designed controlled exper-

Table 5: Comparison between EAG and APG

| Duration | Method | CLAP↑ | IS↑ | FAD↓ | KL↓ |
|---|---|---|---|---|---|
| 30s | CFG | 0.6202 | 9.1372 | 0.9627 | 1.1134 |
| | APG | 0.6201 | 9.1390 | 0.9638 | 1.1107 |
| | EAG | **0.6213** | **9.1514** | **0.9575** | **1.1075** |
| 60s | CFG | **0.6236** | 9.1042 | 0.9707 | 1.1839 |
| | APG | 0.6234 | 9.0937 | 0.9710 | 1.1817 |
| | EAG | 0.6235 | **9.1146** | **0.9630** | **1.1709** |

iments to demonstrate that it effectively combines the advantages of different $\omega$ values rather than performing a naive interpolation.

Table 6: Hyperparameter analysis of EAG

| Duration | $\omega$ | $\delta$ | CLAP↑ | IS↑ | FAD↓ | KL↓ |
|---|---|---|---|---|---|---|
| 30s | 2.5 | - | 0.6235 | 9.1701 | 0.9010 | 1.0713 |
| | 2.5 | 0.8 | **0.6269** | 9.1762 | **0.9008** | 1.0708 |
| | 2.0 | - | 0.6254 | 9.0859 | 0.9086 | 1.0647 |
| | 2.5 | 0.6 | 0.6253 | **9.1819** | 0.9015 | 1.0711 |
| | 1.5 | - | 0.6259 | 8.9013 | 0.9296 | **1.0528** |
| 60s | 2.5 | - | 0.6272 | 9.1096 | **0.9235** | 1.1314 |
| | 2.5 | 0.8 | 0.6279 | **9.1115** | **0.9235** | 1.1255 |
| | 2.0 | - | 0.6260 | 9.0408 | 0.9324 | 1.1136 |
| | 2.5 | 0.6 | **0.6292** | 9.0990 | 0.9247 | 1.1235 |
| | 1.5 | - | 0.6226 | 8.7939 | 0.9554 | **1.0875** |

As shown in *Table 6*, when $\omega = 2.5$ and $\delta = 0.8$, EAG partially emulates the effect of $\omega = 2.0$, yet experimental results show that EAG outperforms both $\omega = 2.0$ and $\omega = 2.5$. Similarly, with $\omega = 2.5$ and $\delta = 0.6$ (approximately equivalent to $\omega = 1.5$), EAG yields better results than either $\omega = 1.5$ or $\omega = 2.5$ alone. These results indicate that EAG intelligently combines the benefits of different guidance strengths in a segment-adaptive manner, leading to more optimal audio generation.

