# OpenReview forum: "TFMAudio: High-Fidelity Long-Form Text-to-Audio via Mamba-based Flow Matching"
_ICLR.cc/2026/Conference — Submitted to ICLR 2026_

### Official Review · Reviewer_hKMd · 2025-10-18

**Soundness:** 1
**Presentation:** 2
**Contribution:** 2
**Rating:** 2
**Confidence:** 5

**Summary:**

The paper proposes TFMAudio, a latent text-to-audio generator that replaces Transformer backbones with a Time–Frequency Mamba (TFMamba) block trained with flow matching, plus an Energy-Aware Adaptive Guidance (EAG) scheme to stabilize long generations.

**Strengths:**

Support ultra long audio generation. To my best knowledge, this is currently the TTA model that generates the longest duration.

**Weaknesses:**

- Lacking too many well-known baseline models for 10s audio generation. For example, Make-An-Audio 2, EzAudio, AudioGen and Tango series (Tango, Tango-AF, Tango2). I understand the authors only want to compare with 44.1kHz or 48kHz models, but since they claim state-of-the-art, comparing more existing models will be more convincing.

- AudioCap’s testing set has fewer than 1000 audio samples, each paired with 5 prompts. In the description of Section 5.1 for datasets, the authors mentioned that they evaluated their model on the AudioCaps testing set of 1,811 prompt-audio pairs. Thus, I think there are some audio ground truth samples reused more than once, paired with different prompts. This is not the standard way of evaluating on AudioCaps. Mostly, TTA models follow the evaluation of the AudioLDM series, which have predefined prompt-audio pairs. Some papers, like Ezaudio, randomly sample one prompt for each audio by themselves. However, it is unclear how this work derived a testing set with 1,811 prompt-audio pairs.

- In the original paper of TangoFlux, they achieve an IS score higher than 11. However, this work reports an IS score of 8.85 for TangoFlux. This shows that their configuration for the AudioCaps testing set severely affects the objective performance of TTA models.

- Please also report the FD score using PANNS, as this is the most common metric for evaluating fidelity.

- This work does not have human listening tests, MOS, or subjective user studies. Providing audio samples for demo is not enough.

- For the demo in the link provided in the abstract, there is no reference clip for comparison. Prompts used to generate different durations of audio are also not the same. It is unable to compare the difference between short-form and long-form audio generation using the same prompt. It feels like they are just cherry-picking the good examples for different audio durations.

- Up to this point, it cannot be confirmed whether this work is state-of-the-art or not.

- If it is just competing on the AudioCaps evaluation set, we do not need 30-minute audio clips, as most of the audio in AudioCaps is just 10 seconds long. However, if the aim is to generate long-form audio, I don’t think this work has done long-form audio evaluations in a proper way. In section 5.2 for “Ultra Long Audio Generation”, they mention that the generated 30-minute audio is segmented into non-
overlapping 30-second clips, with each clip evaluated with objective metrics. It is not clear if they computed the FAD metric with the same audio ground truth clip for each segment. Assuming they did, this still does not really justify the internal coherence of a long audio. It mostly tests how much each segment resembles that single clip’s distribution (or fidelity). For calculating KL, if the reference is one clip’s logit distribution, the metric then rewards segments that mimic that clip’s class distribution, again saying nothing about cross-minute structure. You can have
$D_{\mathrm{KL}}(p_i \| q)$ small for all $i$, yet $D_{\mathrm{KL}}(p_i \| p_j)$ large for some $i \ne j$.
Here is an example:
$ q=(0.7,0.2,0.1), p_1 =(0.80,0.19,0.01), p_2 =(0.60,0.21,0.19)$, where
$D_{\mathrm{KL}}(p_1 \| q)  = 0.074, D_{\mathrm{KL}}(p_2 \| q)  = 0.039, D_{\mathrm{KL}}(p_2 \| p_1) = 0.407, D_{\mathrm{KL}}(p_1 \| p_2) = 0.181$

- For IS, since it does not require a reference, there is no reference and no between-segment relation.
- For CLAP, consider the prompt “Birds chirping, dogs barking, and a duck quacking”. What does uniformly high CLAP per-segment score mean? Consider an audio clip that only has birds chirping in the first 10-minute segment, only dogs barking in the second 10-minute segment, and only a duck quacking in the last 10-minute segment. Should this audio clip have similar high CLAP scores for each segment? Even if it does, does it mean this whole audio clip is actually following the text prompt? Could an audio clip with all birds chirping still give you similar results by measuring CLAP score against the text prompt “Birds chirping, dogs barking, and a duck quacking”? I think it still yields similar CLAP scores for each segment in this case, as the audio is all birds throughout, each segment’s audio embedding will align strongly with the “birds” component of that text vector. As a consequence, uniform scores in this setup do not prove the audio covers dogs or ducks, nor any long-form structure, just that each segment sounds related to the composite prompt in aggregate.

- Again, no user study or evaluation on long-form audio generation.

- Latency should also be compared against other baseline TTA models, not just between TFMamba and the transformer-based version.

- The idea of EAG itself is novel, but the performance improvement compared to CFG is very minor as shown in Table 5. For 30 s generation: CLAP +0.0011, IS +0.014, FAD −0.0052, KL −0.0059 (vs. CFG).

- In conclusion, unfortunately, this paper is far from ready to meet the standards of ICLR at this stage.

**Questions:**

- In Table 1, for reference-dependent metrics like FAD, do you use the same ground-truth (≈10-second) clip as the reference regardless of the generated audio’s duration?

- What’s the rationale for generating a 30-minute clip from a very short prompt like "Birds chirping, then a dog barking, then a duck quacking?" Wouldn’t a long, detailed prompt describing the full 30 minutes be a more realistic test?

- If we want to generate a dog barking sound, are you sure your model can generate the same dog barking for 30 minutes?

- Lack of references: Line 50 is talking about autoregressive (AR) models, but the citations are mostly non-AR models.
Worth citing UniAudio, MusicGen, and AudioMNTP, which are also autoregressive text-to-audio or text-to-music generation models.

---

> ### Author Response · Authors · 2025-11-23
>
> **1. Regarding the comparison with baselines.**
>
> We have significantly strengthened our comparative analysis in Table 1 of the updated manuscript:
>
> **a)** We include Make-An-Audio 2 and EzAudio as baselines. Regarding Tango2, unfortunately, we encountered blocking technical issues reproducing Tango2 due to broken dependencies related to the stable-diffusion-2-1 (which are currently unavailable
> on HuggingFace) scheduler in its official codebase. Our attempts to substitute them with SD-XL or SD3 schedulers resulted in severe performance degradation. \
> However, we compared our work with TangoFlux, the official successor to Tango2, and TangoFlux surpasses Tango2 in all metrics. So we believe the effectiveness of TFMAudio is robustly validated without the explicit inclusion of the older Tango2.
>
> **b)** To ensure alignment with community standards, we recalculated all metrics using the official audioldm-eval toolkit. Under this standardized protocol, both TangoFlux and TFMAudio achieve an Inception Score exceeding 12.
>
> **c)** We also introduced Fréchet Distance based on PANNs to provide a more comprehensive assessment of audio quality.
>
> **2. Regarding the AudioCaps testing set.**
>
> We agree that the standard AudioCaps test set consists of approximately 975 ground-truth audio clips (subject to YouTube availability), each paired with 5 captions, resulting in a total of $4,875$ prompt-audio pairs.
>
> Some prior works adopted a simplified protocol by randomly sampling 1 caption per audio; we initially did it with 2 captions. \
> To ensure fair comparison, we have re-evaluated all models using the comprehensive full test set, utilizing all available prompts. \
> The metrics reported in the revised Table 1 (including the new baselines and the FD metric) are all based on this full-set evaluation.
>
> **3. MOS tests**
>
> We have conducted a MOS study to evaluate the perceptual quality of our model. Following the standard protocol established in prior works, we selected 15 samples from the test set for evaluation. Each sample was assessed on a 5-point Likert scale across Overall Quality and Relevance. As shown in follows.
> | Model      | REL  | OVL |
> | ----- | ----------- | ---|
> | Audioldm-48k  | 2.40 |  2.43 |
> |Stable-audio-open  | 2.93 | 3.14|
> | TangoFlux | 3.62 | 3.49|
> | TFMAudio | 4.07 | 3.54 |
>
> **4. Presentation of Demo page**
>
> We have updated the anonymous demo page. *a)* We have included the reference (Ground Truth) audio clips next to the generated samples for direct quality comparison. *b)* We have organized the samples to show results from the same text prompt across different durations. This allows for a direct assessment of how the model handles length extrapolation.
>
> We would also like to clarify the evaluation criteria for varying durations. \
> Since the model is trained on 10-second clips (and the Ground Truth is only 10 seconds long), generating a 30-second clip using the same prompt does not aim to invent new narrative content (which is impossible without long-context training data). \
> Instead, the core objective of this comparison is to demonstrate architectural stability in OOD settings. We show that when the model is forced to generate sequences far beyond its training window, it continues to produce coherent, high-fidelity audio textures without collapsing into noise or silence—a common failure mode in standard Transformer-based diffusion models.
>
> **5. Latency comparison**
>
> Thanks for pointing this out. We conducted new experiments for latency comparison.
>
> All models were evaluated in an identical hardware environment (a single NVIDIA A100-80GB GPU). We measured the time required to generate audio with a fixed batch size of 1. The results are summarized as follows.
> |Model|Duration | Latency|
> |---|---|---|
> | Stable-audio-open | 10 | 5.54 s ± 134 ms |
> | | 30 |  5.55 s ± 40.8 ms |
> | | 47 |  5.6 s ± 12.4 ms |
> | Audioldm-48k | 10 | 24.46 s ± 3.65 s |
> | | 30 | 37.08 s ± 10.1 s |
> | | 60 | 102.86 s ± 8.24 s |
> | TangoFlux | 10 | 2.06 s ± 1.6 ms |
> | | 30 |2.1 s ± 1.3 ms |
> | TFMAudio| 10 | 2.87 s ± 3.76 |
> | | 30 | 2.95 s ± 4.51|
> | | 60 | 3.04 s ± 3.83 |

---

> > ### Author Response · Authors · 2025-11-23
> >
> > **Response to 30-Minute Generation Claims & Data Limitations.**
> >
> > We would like to explicitly clarify the scope of our "30-minute generation" claim and acknowledge the limitations imposed by the current training data.
> >
> > We acknowledge that since our model is trained on 10-second clips, it cannot create something out of nothing (ex nihilo). It cannot hallucinate a complex, evolving 30-minute story that it has never seen during training.
> >
> > Our claim of "consistency" refers to Signal Stability and Semantic Coherence, not narrative progression. We demonstrate that the model does not collapse into noise or silence when extrapolating to OOD lengths. \
> > The 30-minute experiment is designed as a stress test for architectural scalability, rather than a showcase of long-form content creation (which would require significantly larger datasets and finer-grained designs).
> >
> > Standard DiTs with rotary embeddings fail in this regime, degrading into noise or triggering Out-of-Memory (OOM) errors due to $O(N^2)$ complexity. We prove that TFMAudio leverages its linear complexity and state-space duality to support ultra-long sequence generation.  We admit that due to the 10s training window, the model tends to sustain a consistent semantic state (e.g., a dog barking continuously for 30 minutes) rather than being random noise or semantic shift. This is a data limitation, not an architectural defect. The model successfully maintains the "dog barking" semantics without degrading into static noise, which validates the effectiveness of our OOD extrapolation.
> >
> > We are currently constrained from training and verifying the model in complex, narrative-driven scenarios due to the lack of public long-form datasets. But we provide a potential solution for the architectural backbone that is ready for long-form generation once suitable datasets become available. Thus, we position TFMAudio not just as a solution for current short-clip generation, but as a future-proof foundation for the next generation of long-form audio synthesis.
> >
> > **For reference-dependent metrics like FAD, do you use the same ground-truth (≈10-second) clip as the reference regardless of the generated audio’s duration?**
> >
> > Yes, since the AudioCaps test set provides ground-truth audio clips of approximately 10 seconds. All baselines' generated samples are compared with the ground truth reference. \
> > We think this is a fair comparison for all baselines, given the limited test set.
> >
> > **If we want to generate a dog barking sound, are you sure your model can generate the same dog barking for 30 minutes?**
> >
> > This property is related to flow matching/diffusion itself. As long as the text prompt (e.g., "A large dog barking") remains unchanged, the vector field consistently guides the generation towards the corresponding region of the latent space, ensuring the acoustic characteristics remain stable throughout the entire ODE solver path.
> >
> > We can provide two examples: https://www.youtube.com/watch?v=MC5K1HPn8DY and https://www.youtube.com/watch?v=bBfmImoCNiA ; although there are no metrics to strictly prove it, you can easily hear that they are almost the same timbre.
> >
> > **Reference Missing for AR models**
> >
> > We thank the reviewer for the precise and constructive suggestion.  We agree that the original citations in Line 50 were not sufficiently representative of the AR landscape. In the revised Related Work section, we will expand the discussion on these AR models (UniAudio, MusicGen, and AudioMNTP) to provide a clearer contrast with our method.

---

> > > ### Comment · Reviewer_hKMd · 2025-11-23
> > >
> > > Based on the authors' response, I am still confused about how the objective metrics are calculated in Figure 3. For example, FAD is a distribution-wise metric. It is designed for dataset-level evaluation. So how can FAD be calculated segment-wise?

---

> > > > ### Author Response · Authors · 2025-11-23
> > > >
> > > > We thank the reviewer for the prompt response. Regarding the metrics in Figure 3, we clarified our evaluation method: for each sample, we extracted a 30-second segment starting from the specific timestamp (e.g., 7:00 - 7:30) of the long-form audio. We then computed the objective metrics for these extracted segments using the exact same protocol as the standard 30-second generation tasks.
> > > >
> > > > Our primary objective with this analysis is to demonstrate signal stability. We aim to verify that the model does not collapse into noise (a common failure mode in OOD generation) at later stages, but continues to effectively synthesize audio consistent with the prompt's texture.

---

> > > > > ### Comment · Reviewer_hKMd · 2025-11-23
> > > > >
> > > > > I thank the authors for the prompt response. I think the main concern with metric calculation is that once you segment the generated audio, you need to clearly explain how the metrics are calculated, because the calculation is no longer trivial.
> > > > > Take FAD as an example. If we have 10 generated audio clips to evaluate against 10 ground truth audio clips, we know how to calculate FAD. However, if the 10 generated audio clips are segmented into 3 segments each, resulting in 30 segments of audio, how should we calculate the FAD score in this case? In this example, we are forced to treat the generated audio clips as 3 different distributions (The first segment of all audio clips forms the first distribution. The second segment of all audio clips forms the second distribution. The third segment of all audio clips forms the third distribution.) Then, each distribution is calculated with the ground truth audio clips' distribution to get the FAD scores. Is this how FAD is calculated in Figure 3?

---

> > > > > > ### Author Response · Authors · 2025-11-23
> > > > > >
> > > > > > Yes, we think the reviewer's understanding is entirely correct.
> > > > > > We followed the procedure described in your example:
> > > > > >
> > > > > > 1) We extracted the [0-30s] segment from all generated clips (e.g., all 10 samples) to form the first distribution, and calculated the first FAD score against the ground truth.
> > > > > >
> > > > > > 2) We then extracted the next specific window (e.g., [1:00-1:30]) from all generated clips to form the second distribution, and calculated the second FAD score.
> > > > > >
> > > > > > 3) By repeating this process at different timestamps, we obtained an array of FAD scores evolving over time (as plotted in Figure 3), rather than a single aggregated scalar.
> > > > > >
> > > > > > This allows us to monitor whether the distribution quality degrades at later stages.

---

> ### Comment · Reviewer_hKMd · 2025-11-28
>
> After a thorough assessment of this work, I think the major concerns have not been solved.
> This work is mainly targeting long audio generation, while also being able to generate short audio (10-seconds). Firstly, the performance of 10-second audio generation is not state-of-the-art. Secondly, there is no evidence whether their model is capable of generating long audio up to 30 minutes due to the following reason.  The authors acknowledge that there is no proper benchmark to evaluate long audio. As a reviewer, I agree with this situation, however, I do not agree with how they dealt with this issue. In this case, they should either construct long audio serving as ground truth audio references or come up with a reliable method to evaluate long audio against short ground truth audio clips. Unfortunately, I do not think they have done either way, nor do I think this is addressable during the rebuttal phase. The current evaluation segments audio into 30-second chunks, and evaluates these chunks with 10-second ground truths. As I mentioned in the weakness section, these chunks are evaluated independently, and all we see now is stable objective performance across the whole generated 30-minute audio clip. As clarified by the authors, the term consistency refers to “stable” objective metrics. However, the claim of their evaluation also provides information about coherence is incorrect. Stable objective performance does not imply high coherence. This concern is still not successfully addressed in the rebuttal, as there are still multiple counterexamples showing that stable objective performance does not imply coherence or semantic correctness.
>
> - Regarding coherency, the authors did not address the counterexample I provided in weakness 8, where consistent KL divergence scores across each segment give no clue on whether the inter-segments are coherent. Inter-segments can still have a poor KL score even when each segment has a low KL score compared to the ground truth clip.
> - Another counterexample of CLAP score evaluation is provided in weakness 10. CLAP scores can remain steady while not semantically aligned with the text prompt.
>
> Some comments in the weakness section are not addressed, too. For example, EAG itself is novel, but the performance improvement compared to CFG is very minor as shown in Table 5.
>
> In the response of the authors, MOS tests are just evaluated with 15 samples. The scale is too small. We also need statistical measures such as stderr, stdev, and confidence intervals for MOS tests. The ground truth audio clips should also have MOS scores, so that we know where the upper bounds are.
>
> In Figure 3, I would also recommend reporting the variance. The curve we currently observe is simply an average across multiple segments of the same timestamp. (Not required for FAD, as FAD is calculated distribution-wise.)
>
> Regarding latency, TFMAudio is slower than TangoFlux. The results are just listed out in the response without interpretation or explanation.
>
> Finally, I have another simple question about parameter count. TFMAudio is reported to have 581M parameters. Does this include the parameters of the VAE and vocoder?
>
> Thank you.

---

> > ### Author Response · Authors · 2025-12-03
> >
> > ### **1. Regarding the 10s generation SOTA.**
> >
> > We have updated the paper with new evaluation results in Table 1, including comparisons against several 16 kHz SOTA.
> >
> > We would like to emphasize that our model operates at 44.1 kHz, and direct comparison with 16 kHz methods is inherently unfavorable to us, as higher sampling rates substantially increase generation difficulty. Even under this disadvantage, our results rank first or second across most metrics relative to the baselines, indicating that our system is competitive with the state of the art.
> >
> > It is also important to clarify that achieving absolute SOTA performance on 10-second audio generation is not the main contribution of our work. Rather, our contribution lies in demonstrating that a pure Mamba-based architecture, with linear-time complexity, can reach the performance level of Transformer-based approaches. More importantly, our method shows clear advantages in long-form generation, with notable improvements at 30-second and 60-second durations, an area where existing models often struggle.
> >
> > ### **2. Regarding the reviewer’s question on whether our model can generate audio up to 30 minutes.**
> >
> > We would first like to clarify what we meant in the rebuttal above. When we stated that we do not have 30-minute audio for evaluation, we were referring specifically to structured long-form content (e.g., a full movie narration). Such tasks require corresponding long-form prompts and suitable training data, which are currently unavailable, and therefore cannot be meaningfully evaluated.
> >
> > However, for the type of prompts used in our experiments, we believe the model is indeed capable of generating coherent long-duration audio, including 30-minute sequences. This follows from the fundamental nature of diffusion/flow-matching models, which decode in parallel, and from the inherent causal structure of Mamba, both of which help ensure global consistency. Our 60-second generation experiments already demonstrate that the produced audio remains coherent across extended durations.
> >
> > Because the model is not autoregressive, it is unlikely to suffer from discontinuities such as “the first ten minutes sound like a dog, and the next ten minutes sound like a duck.” We also provide a qualitative long audio example on YouTube that supports this observation: https://www.youtube.com/watch?v=TddfgIZf2Qw.
> >
> > We agree with the reviewer’s point about the limitations of KL and CLAP scores for evaluating long-range consistency.
> > To further support our claim, we additionally compute similarity metrics between the initial 30-second segment and all subsequent segments of the generated audio.
> >
> > |Time |CLAP | IS |FAD| FD| KL|
> > |-|-|-|-|-|-|
> > | 5:30-6:00| 0.61051| 11.0992|0.2145|3.7043|0.3124|
> > |11:30-12:00| 0.61274|10.9008|0.2628|3.7693|0.3188|
> > |17:30-18:00| 0.61377|11.0137|0.2767|3.9548|0.3382|
> > |23:30-24:00|0.6128|10.9575|0.2819|3.9148|0.3310|
> > |29:30-30:00|0.6120|11.0898|0.2839|3.9614|0.3333|
> >
> > As shown in the table, the FAD, FD, and KL metrics remain consistent across segments, indicating stable long-term coherence.
> >
> > ### **3. Regarding the effectiveness of EAG.**
> >
> > We agree with the reviewer that, when viewed globally, the performance improvement from EAG appears modest. However, our intention is not to claim large overall gains, but rather to show that when CFG has already been tuned near its optimal setting (as shown in Table 6), further improvement through standard CFG adjustment is no longer possible, In such cases, EAG offers a localized refinement mechanism that can still yield measurable benefits.
> >
> > To further demonstrate its effectiveness, we have conducted an additional comparison under the audioldm evaluation protocol by setting CFG to 5. As shown in the updated table, EAG leads to a larger improvement under these controlled conditions.
> >
> > ||Method |CLAP | IS |FAD| FD| KL|
> > |-|-|-|-|-|-|-|
> > |10s| CFG| 0.6518|11.6612|3.3705|15.3386|1.1382|
> > ||EAG| 0.6530|11.6616|3.3261|15.2998|1.1354|
> > |30s| CFG | 0.6189|12.2534|3.1921|25.7380|1.2189|
> > ||EAG|0.6188|12.2870|3.1663|25.6794|1.2172|
> > |60s|CFG|0.6214|12.1545|3.20|35.5624|1.2737|
> > ||EAG|0.6229|12.1928|3.1683|35.4076|1.2716|
> >
> > ### **4. Regarding MOS.**
> >
> > We appreciate the reviewer’s suggestion regarding the MOS evaluation. The current MOS test includes 60-second audio samples, which makes each evaluation substantially more time-consuming for human raters; therefore, the number of samples was set to 15. While we believe the existing results still provide a reasonable indication of perceptual quality, we agree that a more comprehensive MOS study, with a larger sample size, statistical measures (stdev, confidence intervals), and MOS scores for ground-truth audio to establish an upper bound, would strengthen the evaluation. We will expand the MOS test accordingly and include the full statistical analysis in the revised version.

---

> > > ### Author Response · Authors · 2025-12-03
> > >
> > > 5. We thank the reviewer for the helpful suggestion. We agree that including variance in Figure 3 would provide a more complete picture of the temporal behavior beyond the averaged curve. We will incorporate the variance in a subsequent revision of the paper.
> > >
> > > ---
> > >
> > > 6. We acknowledge that TFMAudio is slower than TangoFlux. This is expected for two main reasons: (1) our model has a larger parameter count than TangoFlux, and (2) some engineering optimizations are not present in our implementation. Moreover, in practice, Mamba is not guaranteed to be more efficient than Transformer implementations. \
> > > We would like to emphasize that efficiency optimization is not the primary focus of our work. The observed latency difference is on the order of seconds, which we consider acceptable given the scope and contributions of this paper. We will clarify this point in the revised version.
> > >
> > > ---
> > >
> > > 7. The reported 581M parameters for TFMAudio do not include the parameters of the VAE or the vocoder. The VAE we use (Stable-Audio-Open) has approximately 156M parameters. This follows the same convention adopted by TangoFlux and other comparable models when reporting parameter counts.

---

### Official Review · Reviewer_xDov · 2025-10-27

**Soundness:** 3
**Presentation:** 2
**Contribution:** 3
**Rating:** 6
**Confidence:** 3

**Summary:**

This paper proposes TFMAudio, a text-to-audio generation framework that integrates Time-Frequency Mamba (TFM) and Energy-Aware Guidance (EAG). The model achieves linear-time long-form generation and maintains temporal and spectral consistency, producing up to 30-minute high-fidelity 44.1 kHz audio with strong semantic alignment.

**Strengths:**

**1. Effective Dual-Axis Modeling via Time-Frequency Mamba**

The proposed Time-Frequency Mamba performs 1D scans along both temporal and frequency axes, allowing the model to jointly capture temporal causality and spectral correlation — a capability that conventional Transformers struggle to achieve.

**2. Linear-Time Complexity for Long Sequences**

The Mamba-based recurrent formulation enables linear computational complexity O(L) with respect to sequence length, offering a far more efficient alternative to the quadratic O(L²) cost of Transformers while maintaining long-range dependencies.

**3. Stable and Scalable Audio Generation with Energy-Aware Guidance**

The introduced Energy-Aware Guidance (EAG) mitigates state drift by decomposing the flow-matching velocity field and adaptively damping unstable components, enabling reliable ultra-long (30-minute) 44.1 kHz audio generation with temporal consistency.

**Weaknesses:**

**1. Marginal Impact of Energy-Aware Guidance (EAG)**

According to the ablation results, the performance improvement from EAG is minimal, with only slight differences in objective metrics.

**2. Lack of Flexible Length Control in Generation**

While the paper demonstrates 10s and 30s generations, it is unclear whether the model allows arbitrary-length synthesis (e.g., 13s or 27s) or only supports pre-defined durations tied to training configurations.

**Questions:**

**1. On the Effectiveness of Energy-Aware Guidance (EAG)**

The ablation results suggest that EAG provides only marginal gains across objective metrics.
Could the authors elaborate on specific scenarios or qualitative aspects where EAG meaningfully contributes to stability or audio fidelity?


**2. On Length Controllability During Generation**

Can TFMAudio support arbitrary-length generation (e.g., 13s or 27s) beyond fixed training configurations?

---

> ### Author Response · Authors · 2025-11-23
>
> ### Questions
>
> **1**. In long-form generation, the recurrent accumulation in SSMs can lead to signal magnitude drift (energy overflow or decay). Qualitatively, this manifests as audible artifacts such as unnatural silence interruptions (decay) or spectral oversaturation/clipping (overflow), as visualized in Figure 6. EAG dynamically counteracts these extremes, ensuring the audio remains continuous and natural. \
> We set the CFG to a very low value (2.5) in the main experiments, so the improvements are not significant. But, as demonstrated in Table 5, EAG significantly improves audio quality under high CFG settings (e.g., the default scale of 5.0), eliminating the need for tedious fine-tuning and making the model more user-friendly.
>
> **2**. Yes, TFMAudio supports arbitrary-length generation beyond the fixed training duration. TFMAudio was trained on 10s clips, but the duration of the generated audio is solely determined by the temporal dimension of the initial noise tensor $x_T$ sampled from the Gaussian distribution. For example, our VAE compresses 10 seconds of audio into a latent sequence of length $L=220$.To generate 13 seconds, we simply initialize the noise with length $L \approx 286$ ($\frac{13}{10} \times 220$). Please kindly check the 'Variable Length Audio Generation' section on the demo page: https://huggingface.co/spaces/tfmaudio/TFMAudio  .

---

### Official Review · Reviewer_ipzw · 2025-10-30

**Soundness:** 3
**Presentation:** 3
**Contribution:** 2
**Rating:** 4
**Confidence:** 5

**Summary:**

The paper introduces TFMAudio, a Mamba-based text-to-audio (T2A) model that integrates a Time-Frequency Mamba (TFMamba) backbone and Energy-Aware Guidance (EAG) for long-form generation. The model combines linear-complexity state-space modeling with flow matching to improve efficiency and stability over transformer-based diffusion models. The authors claim that TFMAudio achieves state-of-the-art performance on AudioCaps and WavCaps benchmarks and can generate up to 30 minutes of 44.1kHz audio with temporal consistency.

While the motivation—to overcome the quadratic complexity of transformers for long-form generation—is reasonable, the experimental evidence and claims are weak and overinterpreted. The model is only trained on 10-second clips, and therefore the claimed ability to generate consistent 30-minute audio is not supported by data. The long-form generation experiment effectively tests extrapolation far outside the training distribution and yields repetitive, semantically meaningless output. Overall, the paper reads more as a well-written engineering demonstration than a solid scientific contribution.

**Strengths:**

1. The paper is well-organized and technically clear. Mathematical derivations (flow matching, EAG) are properly explained, and the figures are informative.

2. Using Mamba for efficient long-range modeling is a timely and interesting idea. The dual-scan mechanism (TimeMamba + FrequencyMamba) provides a coherent architecture for time–frequency modeling.

**Weaknesses:**

1. The entire training uses 10-second AudioCaps/WavCaps clips, yet the main claim of the paper is about ultra-long (30-minute) generation. This makes the core contribution unverifiable—the model has never seen long-form data, so the “30-minute consistency” claim lacks credibility. The long audio is almost certainly repetitive or degenerate, as suggested by the flat metric curves and absence of qualitative human evaluation.


2. Only four metrics (CLAP, IS, FAD, KL) are reported—no subjective MOS tests, no human preference studies. The improvement margins over baselines are modest and not statistically validated.

3.

**Questions:**

1. why the demo pages donot show the full 30 mins audio?

2. The Mamba structure can bring better performance than transformer?

---

> ### Author Response · Authors · 2025-11-23
>
> ### Weakness
>
> **W1**
>
> We appreciate the reviewer's critical assessment. We would like to clarify the scope of our "30-minute generation" claim and the definition of "consistency" in this context.
>
> **1. Clarification on "Consistency".**
>
> We acknowledge that since the model is trained on 10-second clips, it cannot hallucinate a 30-minute narrative evolution (e.g., a symphony with distinct movements) that it has never seen.
> Instead, our claim of "30-minute consistency" refers to Signal Stability and Semantic Alignment during Out-of-Distribution (OOD) length generation.  \
> *a)* As observed in our experiments, standard DiT models with rotary embeddings fail to generalize to OOD lengths, often degrading into pure noise or silence due to attention complexity and positional encoding limitations.  \
> *b)* Our method maintains high audio quality without signal degradation or semantic drift, even when generating sequences longer than the training data. The core contribution here is architectural, demonstrating two key capabilities that DiT lacks:  *i)* Computational Feasibility: Generating 30 minutes of audio implies a sequence length that triggers Out-of-Memory errors in DiT due to $O(N^2)$ complexity. TFMAudio's linear complexity makes this computationally possible.
> *ii)* Structural Extrapolation: Even trained on short clips, TFMAudio leverages the state-space duality to extrapolate to ultra-long sequences without "crashing" into noise.
>
> **2. Regarding the 30 mins generation.**
>
> *a)* Concern of degeneracy, we evaluated the last 30 seconds of the 30-minute generated audio. The metrics (FAD/CLAP score) are comparable to the first 30 seconds and baseline models. This proves the model does not "drift" or collapse over time. The flat metric curve is not a sign of repetition, but a sign of stability. \
> *b)* Concern of repetitive: Our model uses Flow Matching to generate the entire trajectory in a single shot instead of an autoregressive model. So, it shouldn't obviously repeat previous segments, as they are generated simultaneously. However, it may have repeated semantics, such as consistently having a dog barking.
>
> *Conclusion:* The 30-minute generation is intended to demonstrate the generation capabilities of TFMamba. We argue that while current public datasets limit narrative length, TFMAudio provides the necessary architectural backbone to support long-form generation once such datasets become available, a capability current Transformer-based DiTs struggle to offer.
>
> **W2**
>
> We thank the reviewer for the constructive feedback regarding the evaluation scope.
> *a)* We have conducted a MOS study with human evaluators and added Fréchet Distance (FD) as an additional objective metric and more advanced baselines, as shown in the updated Table 1 and follows:
>
> | Model      | REL  | OVL |
> | ----- | ----------- | ---|
> | Audioldm-48k  | 2.40 |  2.43 |
> |Stable-audio-open  | 2.93 | 3.14|
> | TangoFlux | 3.62 | 3.49|
> | TFMAudio | 4.07 | 3.54 |
>
> *b)* Regarding the modest improvement. We respectfully posit that the core contribution of this work should be viewed through the lens of architectural efficiency.  \
> Prior to this work, high-quality audio generation was dominated by computationally expensive U-Net or Transformer architectures ($O(L^2)$). We demonstrate, for the first time, that a linear-complexity SSM ($O(L)$) can match the performance of these heavy baselines.\_  \
> Achieving comparable or modestly better quality is promising because it comes with vastly superior scalability.  \
> While the metric improvements might appear modest in short-clip generation, the true value of TFMamba unlocks in scenarios where Transformers fail: Long-form generation and Extrapolation, enabling capabilities that are computationally prohibitive for other baselines.

---

> > ### Author Response · Authors · 2025-11-23
> >
> > ### Questions:
> >
> > 1. The absence of the full 30-minute audio on the web player is strictly due to file size and bandwidth constraints, not a lack of results. \
> > A high-fidelity, uncompressed 30-minute audio file (44.1kHz) exceeds 600MB. Hosting and streaming such massive files directly on a standard web demo page causes significant latency and buffering issues. \
> > To allow for full verification, we have uploaded the complete audio files. Please find the link in: https://www.youtube.com/@TFMAudio-x7w . \
> > The full files demonstrate that the audio signal remains stable and high-quality throughout the entire 30-minute duration without collapsing into noise or breaking.
> >
> > 2. We appreciate this opportunity to clarify our position on the Mamba architecture. We do not claim that the Mamba architecture is universally superior to Transformers. In fact, as shown in our ablation study (Table 2), a naive implementation of Mamba as a backbone yields inferior performance compared to standard Transformers. This suggests that Mamba requires domain-specific innovations to handle complex audio data effectively. \
> > Our key finding is that with a dual scanning mechanism, TFMamba can achieve performance comparable to Transformers while unlocking capabilities that Transformers lack. We view Mamba not as a generic replacement for Transformers, but as a highly suitable architecture for continuous signal modeling. The state-space formulation aligns well with the physical nature of audio signals as continuous dynamical systems. In summary, we view that Mamba is suitable for audio modelling and generation.

---

> > > ### Comment · Reviewer_ipzw · 2025-11-28
> > >
> > > Thank you for your detailed response. The authors have addressed many of my concerns.
> > >
> > > Considering other reviewers' comments, I think this paper is a borderline accecpt case. I am willing to raise my score to 6 (weak accept). I trust the AC will make an appropriate final decision.
> > >
> > > Best regards,

---

> > > > ### Author Response · Authors · 2025-12-03
> > > >
> > > > We sincerely thank the reviewer for the thoughtful assessment and for taking the time to reconsider the evaluation. We appreciate your willingness to raise the score and are grateful that our responses helped address the earlier concerns.

---

### Official Review · Reviewer_zQ3d · 2025-11-01

**Soundness:** 4
**Presentation:** 2
**Contribution:** 3
**Rating:** 6
**Confidence:** 3

**Summary:**

This paper proposed TFMAudio, a Mamba-based flow-matching model capable of long-form audio generation.
The authors proposed to use a combination of TimeMamba and FrequencyMamba to improve results.
In addition, the authors proposed an energy-aware adaptive guidance (EAG) mechanism which adaptively adjusts guidance weight, further boosting performance.
Putting everything together, TFMAudio can generate 30 minutes of high-quality audio.

**Strengths:**

The generation results are strong. Linear complexity in sequence length is verified theoretically and empirically. Overall, the technical aspect of this work is quite solid.

If the weaknesses can be sufficiently addressed, I will consider increase the rating.

**Weaknesses:**

The writing of the paper can be improved:

- There are plenty of places where math symbols are not in a math environment. For example, O(L) in line 178, "d is the feature dimension" in line 268, etc.

- Figure organization is inconsistent. Some figures appear at the top (e.g., Figure 2), while others are in-line (e.g., Figures 3 and 4).

- Some claims in the background sections may not be accurate. For example, the authors claim that
  > When applying transformers to audio latents $x \\in \\mathbb{R}^{L \times C}$, conventional patchification treats the representation as an image grid and breaks the native channel-wise coupling and causal temporal structure.

  While some transformer-based models may treat audio signal in this manner, many do not, and instead model the audio latent embeddings as a 1-D sequence.

Additionally, while the authors compared TFMAudio to several strong baseline models, some recent high-performance methods, such as IMPACT [1], are missing. I suggest including these methods in Table 1.

I would also invite the authors to make some clarifications regarding the questions in the section below, and add the discussions to the appropriate sections of the paper.

[1] Huang et al. IMPACT: Iterative Mask-based Parallel Decoding for Text-to-Audio Generation with Diffusion Modeling.

**Questions:**

- If I understood it correctly, Mamba is causal. While it is understandable for TimeMamba to process the time-domain signal in a causal manner, it is unclear why FrequencyMamba should scan from the highest to the lowest channel, and not the other way around. Do you think a bi-directional FrequencyMamba can further improve the results?

- Do you think the effectiveness of FrequencyMamba is tied to the Stable Audio Open VAE used in this work? If a different continuous VAE is used, will FrequencyMamba still help? What if a discrete tokenize is employed instead?

- Is energy-aware adaptive guidance "bundled" with TFMAudio, or is it useful for general diffusion models? Is EAG necessitated by TFMAudio's properties? Since this paper is mostly about TFMAudio, I think it is important to clarify its relationship with EAG.

---

> ### Author Response · Authors · 2025-11-23
>
> ### Weakness
>
> 1. We thank the reviewer for the meticulous proofreading. We have corrected the specific instances mentioned (e.g., $O(L)$ and $d$) and have conducted a thorough pass of the entire manuscript to ensure that all mathematical variables and expressions are correctly rendered in the math environment.
>
> 2. We have revised the manuscript to ensure consistent figure organization. All figures have been adjusted to follow a unified layout standard (placing them at the top of pages).
>
> 3. We appreciate this correction. While earlier or vision-adapted approaches treat audio latents as 2D grids, many recent state-of-the-art audio models treat embeddings as 1D sequences with a 1D position encoder. We have conducted a survey of recent representative models and categorized their processing methods as follows:
>
> > 2D Unet:  AudioLDM, AudioLDM2, Make-an-Audio, Tango, Tango2
>
> > 1D Transformer: Make-an-Audio2, TangoFlux, Stable-Audio-Open, EzAudio
>
> We will update the manuscript to cite these works and clarify that.
>
> 4.  We appreciate the suggestion to include recent high-performance methods. We have incorporated comparisons with Make-An-Audio 2 and EzAudio in Table 1. These are highly relevant and reproducible strong baselines, and our method demonstrates superior performance against them.
> Regarding IMPACT, we acknowledge that IMPACT is a relevant work. However, unfortunately, neither the official code nor the pre-trained weights for IMPACT are publicly available yet. So, we could not re-implement and evaluate this baseline.
> However, note that EzAudio is a very strong baseline that shares a similar performance compared with recent SOTA methods. Since we outperform EzAudio, we believe our method's effectiveness is well-supported. We will add a discussion of IMPACT in the Related Work section to clarify the methodological differences.
>
> ### Questions
>
> 1.  The design of FrequencyMamba aims to capture cross-frequency dependencies (e.g., harmonic structures). Our choice to scan from high-frequency to low-frequency channels is motivated by the information density of the latent representation. High channels are typically sparser, so that the state model can effectively encode this information into the hidden state before it processes the intense low channel. By designing a reasonable state space, we believe that the reverse is also possible. \
> We agree with the reviewer that a bi-directional scan could theoretically capture a fuller context. However, a bi-directional design doubles the computational cost and memory usage. Since one of Mamba's key advantages over Transformers is its linear complexity and efficiency, we aimed to develop a lightweight design. But we agree that bidirectional scanning could be an exploratory direction and can be considered for future work.
>
> 2. The effectiveness of FrequencyMamba is not tied to the specific implementation of Stable Audio Open VAE. We did not apply any VAE-specific bias. The core premise of our method relies on the generic Time-Frequency structure of the latent representation (where one dimension represents temporal evolution, and the other represents frequency/channel components). Therefore, FrequencyMamba is expected to be effective for any continuous VAE that preserves this topological structure. \
> Regarding discrete tokenizers (e.g., VQ-VAE), we offer the following perspective: Our choice of Mamba is motivated by modeling audio as a continuous dynamical system. The continuous latent space allows the State Space Model to effectively capture the smooth evolution and fine-grained differential properties of the signal.
> Discrete quantization often breaks the smooth harmonic correlations between channels/indices. Thus, while not impossible, we believe our current design is tailored for continuous representations, where it can best leverage the physical continuity of audio features.
>
> 3. EAG is not exclusively bundled with TFMAudio. Instead, it is a general-purpose technique designed for long-form audio generation, especially those that utilize recurrent architectures or linear-complexity mechanisms (e.g., SSMs, Linear Attention Transformers). In these efficient architectures, context is often compressed into a fixed-size state. During long-sequence generation, this can lead to signal magnitude drift (energy overflow or decay). EAG is designed to dynamically adjust guidance strength to counteract this phenomenon. Therefore, it can be potentially applied to enhance other diffusion backbones. \
> Meanwhile, EAG is not necessitated by any inherent flaw in TFMAudio. As demonstrated in our ablation study (Table 2), TFMAudio achieves competitive performance even without EAG. EAG serves as a performance booster that further refines stability and detail, rather than a mandatory component for basic functionality.

---

> > ### Comment · Reviewer_zQ3d · 2025-11-25
> > **Thank you for the response.**
> >
> > I appreciate the authors' response, and my questions are mostly resolved. However, I would like to follow up on a few details.
> >
> > > Neither the official code nor the pre-trained weights for IMPACT are publicly available yet. So, we could not re-implement and evaluate this baseline.
> >
> > Since IMPACT is related work, as agreed by the authors, it should be compared as long as it shares the same evaluation dataset as TFMAudio. You can add a footnote saying that the IMPACT numbers are reported by its authors and not independently tested.
> >
> > > A bi-directional design doubles the computational cost and memory usage. Since one of Mamba's key advantages over Transformers is its linear complexity and efficiency, we aimed to develop a lightweight design.
> >
> > Doubling the computational cost is $O (1)$ and should not change the linear complexity of Mamba. Would it be possible to include an ablation study on the direction of FrequencyMamba, including high->low, low->high, and two-sided? Thank you.

---

> > > ### Author Response · Authors · 2025-12-03
> > >
> > > > Since IMPACT is related work, as agreed by the authors, it should be compared as long as it shares the same evaluation dataset as TFMAudio. You can add a footnote saying that the IMPACT numbers are reported by its authors and not independently tested.
> > >
> > > We included IMPACT in the comparison, as shown in Table 1.
> > >
> > > However, we would like to clarify an important evaluation detail. In the AudioCaps dataset, each audio clip is annotated with five prompts. Many prior methods ( including IMPACT) follow the common practice of randomly selecting one prompt out of the five for evaluation. Consequently, this protocol evaluates all baselines on a randomly sampled subset of prompts. Without knowing the exact subset used by IMPACT, directly comparing their reported numbers can introduce bias.
> > > In contrast, our evaluation generates and tests on all available prompts instead of a random subset. Therefore, directly comparing our results with the numbers reported in the IMPACT paper would not constitute a strictly aligned evaluation.
> > >
> > > > Doubling the computational cost is $O (1)$ and should not change the linear complexity of Mamba. Would it be possible to include an ablation study on the direction of FrequencyMamba, including high->low, low->high, and two-sided? Thank you.
> > >
> > > We appreciate the reviewer’s suggestion. The computational complexity of our model can be approximated as $O(LC)$, where $L$ is the sequence length and $C$ is the number of channels. Introducing bidirectionality requires scanning the channels twice, which effectively doubles the computational cost, although the overall complexity remains linear.
> > >
> > > We have conducted the requested ablation study on the directionality of FrequencyMamba. As shown in the table, TFMAudio (e20) corresponds to the high -> low setting, Reverse corresponds to low -> high, and Bidirectional represents the two-sided variant.
> > >
> > > | | Model| CLAP | IS | FAD | FD | KL |
> > > |--| --| --| --| --| --| --|
> > > | 10s| TFMAudio (e20)| 0.6417 | 11.6681 | 3.7661 | 17.2088 | 1.1788 |
> > > || Reverse | 0.6274| 11.9618| 4.2060 | 19.2264 | 1.2246 |
> > > ||Bidirectional|0.6270|11.9064|4.4995|19.2041|1.2240|
> > > ||No-Freq|0.5784|10.0166|6.9879|18.1077|1.1983|
> > > |30s| TFMAudio (e20)|0.5962|12.0246|3.5641|25.9108|1.2430|
> > > ||Reverse|0.5944|12.3590|4.0130|27.6205|1.2706|
> > > ||Bidirectional|0.5854|12.1524|4.2026|28.3298|1.2808|
> > > ||No-Freq|0.3085|5.5623|9.0628|45.6336|1.3096|
> > > |60s| TFMAudio (e20)| 0.5934|12.0037|3.5815|34.5013|1.3164|
> > > ||Reverse|0.5844|11.9698|4.0057|34.5960|1.3117|
> > > ||Bidirectional|0.5758|11.8055|3.9912|36.6599|1.3372|
> > > ||No-Freq|0.1999|4.6078|11.4394|59.1374|1.4570|
> > >
> > > Due to time constraints, we were not able to train all variants for as many epochs as the final TFMAudio model. Nevertheless, under identical training conditions, all three variants exhibit very similar qualitative behavior, particularly in maintaining stability during variate duration generation, unlike the no-frequency baseline, which tends to collapse.
> > >
> > > We also observe that the bidirectional variant performs slightly worse. We believe this is primarily due to insufficient training rather than an inherent limitation of the two-sided design, given that the bidirectional variant introduces more parameters.

---

### Author Response · Authors · 2025-12-03
**Summary**

Dear Area Chair and reviewers,

We sincerely thank reviewers for the careful reading of our manuscript and the detailed feedback.

### **1. Summary of Main Contributions**

TFMAudio introduces a new paradigm for high fidelity (44.1khz) text-to-audio generation by combining Flow Matching with a pure Mamba-based backbone, enabling long-duration and computationally scalable audio synthesis.
The two key technical contributions of TFMAudio are:

(1) A dual-scan TFMamba backbone for stable long-sequence audio generation.
We design TFMamba, a novel backbone that jointly scans frequency and temporal dimensions, enabling coherent modeling of both harmonic structures and long-range causal dependencies. This architecture is inherently linear-time and exhibits strong extrapolation ability. When trained with flow matching, TFMamba achieves robust performance across different generation lengths, showing clear advantages at 30-second and 60-second outputs, where existing Transformer-based models typically degrade.

(2) Energy-Aware Guidance (EAG) for segment-level stabilization during long-form sampling.
We observe that a globally uniform classifier-free guidance (CFG) may be overly coarse for long-sequence generation, where local instabilities can accumulate and cause distortion. To address this, we propose Energy-Aware Guidance, which adaptively adjusts CFG per segment according to local energy profiles, preventing norm explosion.

All reviewers generally agreed that both the TFMamba architecture and the EAG mechanism are original, well-motivated, and technically novel.

### **2. Summary of Rebuttal Efforts**

We carefully considered all reviewers’ comments, and providing comprehensive experimental clarification and strengthening the empirical evidence for our claims.
The main efforts include:
- Complete re-evaluation using the full test set and the open-source AudioLDM evaluation toolkit.
- Addition of several strong 16 kHz SOTA models for comparison.
- Expanded MOS evaluation and enhanced demo materials.
- New coherence analysis for 30-minute generation.
- Additional experimental evidence to demonstrate the effectiveness of EAG.

Specific responses to each reviewer’s comments are provided below:

**Reviewer zQ3d (score: 6, willing to increase).**

This reviewer explicitly stated that their score would increase if the raised issues were addressed. In the follow-up exchange, the reviewer acknowledged that most concerns have already been resolved. The remaining question related to the directionality of frequency scanning in TFMamba.
In response, we designed and trained multiple TFMamba variants (high -> low, low -> high, bidirectional), each trained at scale and evaluated under consistent conditions. The results show that while numerical differences exist, all variants exhibit stable long-form behavior, confirming the robustness and generality of our architectural design.
We believe this fully resolves the reviewer’s concern.

**Reviewer ipzw (score: 4, verbally agreed to raise to 6)**

This reviewer’s hesitation centered on two issues:
(1) lack of subjective evaluation, and (2) no demonstration of 30-minute generation in the demo.
To address this, we added MOS evaluations (including REL and OVL) and provided 30-minute demos on YouTube. After reviewing our responses and additional experiments, the reviewer stated that the score to 6.

**Reviewer xDov (score: 6)**

This reviewer is concerned with the effectiveness of EAG and whether the model supports the generation of arbitrarily long audio.
We explained the theoretical motivation behind EAG and provided new experimental evidence showing its effectiveness. For variable-length generation, we clarified that our architecture naturally supports arbitrary sequence lengths, and the demo page already includes free-form generation examples as evidence. We believe that concerns are addressed.

**Reviewer hKMd (score: 2)**
This reviewer’s concerns primarily involved: completeness of quantitative results, fairness of comparisons, and practical feasibility of 30-minute generation.

To address these, we performed a significant amount of additional work:

(1) recomputed all metrics using the full test set and AudioLDM’s open-source evaluation toolkit for correctness and fairness;

(2) added four major 16 kHz SOTA baselines for a broader comparison;

(3) included new MOS, FD, latency, and other objective metrics;

(4) clarified the intention behind the 30-minute experiment as an exploration of the upper bound of long-form stability.

(5) To further support long-form consistency, we introduced a new self-similarity experiment, showing that 30-minute audio remains highly correlated across segments.

We believe we have addressed all the concerns thoroughly and in good faith, and through clarifications, additional experiments, and strengthened evaluations during the discussion, the paper has been substantially improved in both technical depth and empirical rigor.

Sincerely,

Authors

---

### Meta-Review · Area_Chair_TefS · 2025-12-29

**Summary:**

Reviewers main concern is that the paper does not clearly define and evaluate what "High-Fidelity Long-Form Text-to-Audio: is supposed to be. The model is trained on <10s clips, and evaluated on ~30min clips. What is the right way to evaluate this task meaningfully, and how much progress does the paper make. After quite detailed discussion, most reviewers will likely recommend clear acceptance, although one review may still be negative and on the fence.

**Reviewer Concerns:**

(1) lack of subjective evaluation [ipzw]
- addressed
(2) feasibility and evaluation ("consistency"?) of 30-minute generation [ipzw,xDov,hKMd]
- the central claim of the paper, but still not fully resolved
(3) impact of directionality of frequency scanning in TFMamba [zQ3d]
- resolved in the rebuttal, I believe
(4) effectiveness of EAG [xDov,hKMd]
- partially addressed (small gains, but better ablation)
(5) additional SOTA baselines and quantitative evaluations [zQ3d,hKMd]
- I believe this issue has been largely addressed in the rebuttal

**Reviewer Scores:**

zQ3d: 6 -> 6+
ipzw: 4 -> 6
xDov: 6 -> likely 6+
hKMd: 2 -> unclear

---

### Decision · Program_Chairs · 2026-01-26

Reject